# GSmooth: Certified Robustness against Semantic Transformations via Generalized Randomized Smoothing

## Abstract

The vulnerability of deep learning models to adversarial examples and semantic transformations has limited the applications in risk-sensitive areas. The recent development of certified defense approaches like randomized smoothing provides a promising direction towards building reliable machine learning systems. However, current certified defenses cannot handle complex semantic transformations like rotational blur and defocus blur which are common in practical applications. In this paper, we propose a generalized randomized smoothing framework (GSmooth) for certified robustness against semantic transformations. We provide both a unified and rigorous theoretical framework and scalable algorithms for certified robustness on complex semantic transformations. Specifically, our key idea is to use a surrogate image-to-image neural network to approximate a transformation which provides a powerful tool for studying the properties of semantic transformations and certify the transformation based on this neural network. Experiments on multiple types of semantic perturbations and corruptions using multiple datasets demonstrate the effectiveness of our approach.

## 1 Introduction

Although deep learning models have achieved remarkable success on various applications (LeCun et al., 2015), they are vulnerable to adversarial examples (Biggio et al., 2013; Szegedy et al., 2013; Goodfellow et al., 2014) and semantic transformations (Hendrycks & Dietterich, 2019). The vulnerability of deep learning models can limit their applications on many important tasks. For example, the autonomous driving system can be misled even by a small adversarial patch on the road mark (Jing et al., 2021). Compared with the maliciously crafted adversarial examples, semantic transformations are more practical in real-world scenarios, such as rotation, translation, blur, bad weather, and so on. Such transformations do not damage the semantic features of images and can be easily recognized by humans, but they also degrade the performance of deep learning models. Therefore, it is imperative to improve model robustness against semantic transformations.

To develop more reliable machine learning systems, many efforts have been made to design defense techniques against adversarial attacks or semantic transformations. The existing defense methods can be categorized into empirical defenses and certified defenses. Adversarial training (AT) (Madry et al., 2017; Zhang et al., 2019a) is one of the most effective empirical defenses against $\ell_p$-norm bounded adversarial examples. Moreover, methods based on data augmentation (Hendrycks et al., 2019; Wang et al., 2019; Calian et al., 2021) have been proposed to empirically improve the performance under semantic transformations. However, the performance of empirical defenses is difficult to be fully justified and these defenses can be further broken by new adaptive attacks (Athalye et al., 2018; Tramer et al., 2020). In contrast, the certified defenses aim to theoretically provide a certified region where the model is theoretically safe under any attack or perturbation (Wong & Kolter, 2018; Cohen et al., 2019; Gowal et al., 2018; Zhang et al., 2019b). Along this line, developing certified defense methods is a crucial step towards reliable machine learning systems.

Although certified defenses have achieved great success, most of them are limited to defend against $\ell_p$-norm bounded attacks. However, the $\ell_p$ distance between the original image and its corrupted counterpart by a semantic transformation (e.g., translation, rotation) would be large even when the

corruption is slight. Therefore, the current methods are incapable of certifying robustness against such semantic perturbations. To solve this problem, several recent works (Fischer et al., 2020; Mohapatra et al., 2020; Li et al., 2021) attempt to extend the certified defenses to several simple semantic corruptions, including translation, rotation, and Gaussian blur. However, these works are not scalable to certify robustness against complex and general semantic perturbations. First, deterministic certified defenses (Mohapatra et al., 2020) based on convex relaxation for the activation function require solving a complex optimization problem for computing bound which is computationally expensive. Second, probabilistic approaches based on randomized smoothing also demand a handcrafted Lipschitz bound (Li et al., 2021), which is intractable for complicated semantic transformations. For example, many semantic transformations such as glass blur and pixelate do not have a closed form expression or they are black boxes and hard to be analyzed theoretically, but they are common in real-world scenarios. Therefore, it is still highly challenging to certify robustness against these complex and realistic semantic transformations.

To address the aforementioned challenges, we propose a *generalized randomized smoothing* framework (GSmooth). First, we provide a unified framework of GSmooth for certifying general semantic transformations. Then we categorize the transformations into resolvable transformations (e.g., translation) and non-resolvable transformations (e.g., rotational blur) similar with Li et al. (2021). As mentioned above, most non-resolvable transformations are complex and the existing methods cannot provide their certified radius. To handle the challenge, we propose to use an image-to-image translation neural network to approximate all these transformations. Due to the strong capacity of neural networks, our method is flexible and scalable to model these complex semantic transformations. By introducing an augmented noise in the layers of the surrogate model, we can theoretically provide the certified radius for the proxy neural networks which can be used for certifying the original transformations. Next, we provide theoretical analysis and error bounding for the approximation. Finally, we validate the effectiveness of our methods on several publicly available datasets. Extensive experimental results demonstrate that our methods are effective for certifying complex semantic transformations including different types of blur or image quality corruptions.

## 2 RELATED WORK

### 2.1 ATTACKS AND DEFENSES FOR SEMANTIC TRANSFORMATIONS

Unlike $\ell_p$ perturbation which adds small noise to every pixel of an image, semantic attacks or physical attacks are usually unrestricted. Brown et al. (2017); Song et al. (2018) use a small patch added to the image to mislead the classifier or the object detector. Engstrom et al. (2019; 2018); Xiao et al. (2018) construct adversarial examples using spatial transformations like rotation or translation. Hendrycks & Dietterich (2019) show that a wide variety of semantic perturbations degrade the performance for many deep learning models. Many works (Cubuk et al., 2019; Hendrycks et al., 2019; 2020; Robey et al., 2020) propose diverse data augmentation techniques to enhance robustness under semantic perturbations. Calian et al. (2021) propose adversarial data augmentation that can be viewed as adversarial training for defending semantic perturbations. Beyond empirical defenses, several works (Mohapatra et al., 2020; Madry et al., 2017; Singh et al., 2019; Balunović et al., 2019) attempt to certify some simple geometric transformations. However, all of them belong to deterministic certification approaches and their performance on realistic datasets are unsatisfactory.

### 2.2 RANDOMIZED SMOOTHING

Randomized smoothing is a novel certification method originated from differential privacy (Lecuyer et al., 2019). Cohen et al. (2019) then improve the certified bound and apply it to large scale deep neural networks and datasets. Yang et al. (2020) exhaustively analyze the robust radius by using different noise distribution and norms. Hayes (2020); Yang et al. (2020) point out that randomized smoothing suffers from curse of dimensionality for the $l_\infty$ norm. Salman et al. (2019) adopt adversarial training to train smoothed classifiers to obtain better robustness guarantees. Li et al. (2021); Fischer et al. (2020) extend randomized smoothing to certify some simple semantic transformations, e.g., image translation and rotation. It shows that randomized smoothing could be generalized to certify more diverse attacks or corruptions. However, their methods are limited to simple semantic transformations, which are easy to analyze their mathematical properties.

## 3 PROPOSED METHOD

In this section, we present the framework and theoretical analyses of our Generalized Randomized Smoothing (GSmooth). We first introduce the basic notations. Then we divide the semantic transformations into resolvable transformations and non-resolvable transformations similar with Li et al. (2021). Next we introduce the details of our GSmooth for these two types of semantic transformations, respectively. Finally, we show the theoretical insight and proof sketch of our main results.

### 3.1 NOTATIONS

We first introduce the notations and formulation of the task. Given the input of $x \in \mathbb{R}^n$ and the labels of $\mathcal{Y} = \{1, 2, \ldots p\}$, we denote the classifier as $f(x) : \mathbb{R}^n \to [0, 1]^p$, which outputs predicted probabilities over all $p$ classes. The prediction of $f$ is $\arg \max_{i \in \mathcal{Y}} f(x)_i$, where $f(\cdot)_i$ denotes the $i$-th element of $f(\cdot)$. Let $\tau(\theta, x) : \mathbb{R}^m \times \mathbb{R}^n \to \mathbb{R}^n$ be a semantic transformation of raw input $x$ with parameter $\theta \in \mathbb{R}^m$. We define the smoothed classifier as

$$G(x) = \mathbb{E}_{\theta \sim g}[f(\tau(\theta, x))], \tag{1}$$

which is the average prediction for the samples under a smoothing distribution $g(\theta)$ where $g(\theta) = \exp(-\psi(\theta))$ and $\psi(\theta)$ is a smooth function from $\mathbb{R}^m \to \mathbb{R}$. Let $||u|| = 1$ be any vector with unit norm and a random variable $\gamma_u = \langle u, \nabla \psi(\delta) \rangle$ where $\delta \sim g$ and $\nabla$ is the gradient operator of a function. The complementary CDF is $\varphi_u(c) = \mathbb{P}[\gamma_u > c]$ and the inverse complementary CDF is $\varphi_u^{-1}(p) = \inf\{c | \mathbb{P}(\gamma_u > c) \leqslant p\}$. Following Yang et al. (2020), we define a function $\Phi$ as

$$\Phi(p) = \max_{||u||=1} \mathbb{E}[\gamma_u \mathbb{I}\{\gamma_u > \varphi_u^{-1}(p)\}], \tag{2}$$

which will be used to represent the certified radius. Let $y_A = \arg \max_{i \in \mathcal{Y}} G(x)_i$ be the predicted label by the smoothed classifier $G(x)$ and $y_B = \arg \max_{i \in \mathcal{Y} \setminus y_A} G(x)_i$ is the runner-up class. Without causing confusion, we use $G(x)_A$ to denote the probability of the top class $G(x)_{y_A}$; likewise for $G(x)_B$.

### 3.2 CERTIFIED BOUND FOR RESOLVABLE SEMANTIC TRANSFORMATIONS

We first discuss a class of transformations that are resolvable — the composition of two transformations with parameters belonging to a perturbation set $\theta, \xi \in P \subset \mathbb{R}^m$ is still a transformation with a new parameter $\gamma(\theta, \xi) \in P \subset \mathbb{R}^m$, where $\gamma(\cdot, \cdot) : P \times P \to P$ is a function depending on these parameters. For resolvable semantic transformations, we have the following theorem.

**Theorem 1.** *Let $f(x)$ be any classifier and $G(x)$ be the smoothed classifier defined in Eq. (1). If there exists a function $M(\cdot, \cdot) : P \times P \to \mathbb{R}$, the transformation $\tau(\cdot, \cdot)$ satisfies*

$$\frac{\partial \gamma(\theta, \xi)}{\partial \xi} = \frac{\partial \gamma(\theta, \xi)}{\partial \theta} M(\theta, \xi),$$

*and there exist two constants $\underline{p_A}$ , $\overline{p_B}$ satisfying*

$$G(x)_A \geqslant \underline{p_A} \geqslant \overline{p_B} \geqslant G(x)_B,$$

*then $y_A = \arg \max_{i \in \mathcal{Y}} G(\tau(\xi, x))_i$ holds for any $||\xi|| \leqslant R$ where*

$$R = \frac{1}{2M^*} \int_{\overline{p_B}}^{\underline{p_A}} \frac{1}{\Phi(p)} dp, \tag{3}$$

*and $M^* = max_{\xi, \theta \in P} ||M(\xi, \theta)||$.*

**Remark.** The settings of Theorem 1 are similar with Li et al. (2021) for resolvable semantic transformations. But here we adopt a different presentation and proof for the theorem which could be easier to extend to our GSmooth framework for general semantic transformations. Specifically, we show two examples of the theorem which are additive transformations and commutable transformations. A transformation is additive if $\tau(\theta, \tau(\xi, x)) = \tau(\xi + \theta, x)$ for any $\theta, \xi \in P$; or it is commutable if $\tau(\theta, \tau(\xi, x)) = \tau(\xi, \tau(\theta, x))$ for any $\theta, \xi \in P$. For these two types of transformations, it is straightforward to verify that they satisfy the property proposed in Theorem 1. As an example, we simply apply Theorem 1 for isotropic Gaussian distribution $g(\theta) = \mathcal{N}(0, \sigma^2 I)$ and get the certified radius

$$R = \frac{\sigma}{2} \left( \Psi \left( \underline{p_A} \right) - \Psi \left( \overline{p_B} \right) \right), \tag{4}$$

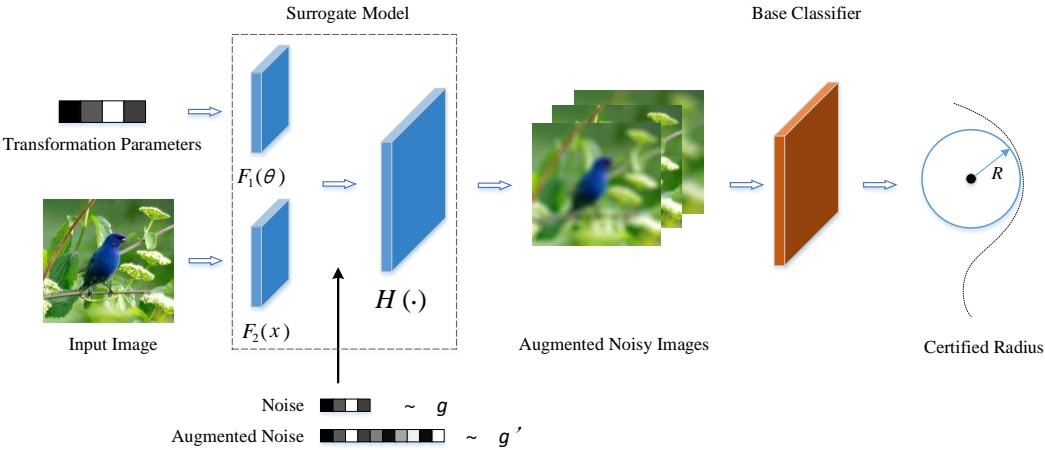

Figure 1: A graphical illustration of our GSmooth. We use a surrogate image-to-image translation network to accurately fit a semantic transformation. Then we add a new augmented noise into the surrogate model and construct the GSmooth classifier. The augmented noise are sampled to ensure the transformation to be resolvable in the semantic space. We theoretically calculate the certification bound for the surrogate model to certify the original semantic transformation.

where $\Psi$ is the inverse CDF of the standard Gaussian distribution. These two kinds of transformations include image translation and Gaussian blur, which are basic semantic transformations and widely discussed in previous works (Li et al., 2021; Fischer et al., 2020). The certification of these simple transformations only requires applying translation or Gaussian blur to the sample and gets the average classification score under the noise distribution.

### 3.3 CERTIFIED BOUND FOR GENERAL SEMANTIC TRANSFORMATIONS

Translation and Gaussian blur are two specific cases of semantic transformations. In practice, most semantic transformations are not commutable or even not resolvable. Therefore, we need to develop better methods for certifying more types of semantic transformations. However, the existing methods like Semanify-NN (Mohapatra et al., 2020) based on convex relaxation and TSS (Li et al., 2021) based on randomized smoothing require to develop a specific algorithm or bound for each individual semantic transformation. This is not scalable and might be infeasible for more complicated transformations without explicit mathematical forms.

To address the challenge, we draw inspiration from the fact that neural networks are able to approximate functions including a complex and unknown semantic transformation (Zhu et al., 2017). First, we propose to use a surrogate image-to-image translation model to accurately fit the semantic transformation. Then we theoretically show that by introducing an augmented noise in the layers of the surrogate model, as shown in Fig 1, randomized smoothing can be extended to handle these transformations. Specifically, we define the surrogate model as the following form that will lead to a simple certification bound as we shall see:

$$\tau(\theta, x) = H(F_1(\theta) + F_2(x)), \tag{5}$$

where $F_1(\cdot) : \mathbb{R}^m \to \mathbb{R}^d, F_2(\cdot) : \mathbb{R}^n \to \mathbb{R}^d$, and $H(\cdot) : \mathbb{R}^d \to \mathbb{R}^n$ are three individual neural networks. $F_1(\cdot)$ and $F_2(\cdot)$ are the encoders for transformation parameters and images respectively, and their encodings are added together in the semantic space which is critical for its theoretical certification. We find that the surrogate neural network is much easier to analyze and can be certified by introducing a dimensional augmentation strategy for both noise parameters and input images.

As illustrated in Fig. 1, an augmented noise is added to the semantic layers $H(\cdot)$ in the surrogate model. Our key insight is that the transformation could be viewed as the superposition of a resolvable part and a non-resolvable residual part in the augmented semantic space. Then we could use the augmented noise to control the non-resolvable residual part if the augmented dimension $d \geqslant m + n$. This dimension augmentation is the key step of our technique. The augmentation for noise is from $\mathbb{R}^m$ to $\mathbb{R}^d$. To keep the dimension consistent, we also augment data $x$ to $\mathbb{R}^d$ by padding 0 entries.

By certifying the transformation based on the surrogate model, we are able to certify the original transformation if the approximation error is within an acceptable region (detailed analysis is in Theorem 3). Our method is flexible and scalable since the surrogate neural network has a uniform mathematical form for theoretical analysis and they are trained automatically. Next, we discuss the details of GSmooth.

Specifically, we introduce the augmented data $\tilde{x} \in \mathbb{R}^d$ and the augmented parameter $\tilde{\theta} \in \mathbb{R}^d$ as

$$\tilde{x} = \begin{pmatrix} x' \\ x \end{pmatrix}, \tilde{\theta} = \begin{pmatrix} \theta \\ \theta' \end{pmatrix}, \tag{6}$$

where the additional parameters $\theta' \in \mathbb{R}^n$ are sampled from $g'(\theta')$, and the joint distribution of $\theta'$ and $\theta$ is $\tilde{\theta} \sim \tilde{g}$ where $\tilde{g}(\tilde{\theta}) = g'(\theta')g(\theta)$. Moreover, the augmented data $x' \in \mathbb{R}^m$. We define the *generalized smoothed classifier* as

$$\tilde{G}(\tilde{x}) = \mathbb{E}_{\tilde{\theta} \sim \tilde{g}(\tilde{\theta})} \left[ \tilde{f}(\tilde{\tau}(\tilde{\theta}, \tilde{x})) \right], \tag{7}$$

where $\tilde{f}$ is the "augmented target classifier" that equals the original classifier when constrained on the original input $x$, which means $\tilde{f}(\tilde{x}) = f(x)$. This can be achieved by setting the weights of additional dimensions to 0. Note that now all the functions are augmented for a $d$ dimensional input. We further augment our surrogate neural network to represent the augmented transformation $\tilde{\tau} : \mathbb{R}^d \to \mathbb{R}^d$,

$$\tilde{\tau}(\tilde{\theta}, \tilde{x}) = \tilde{H}(\tilde{F}_1(\tilde{\theta}) + \tilde{F}_2(\tilde{x})), \tag{8}$$

where $\tilde{H}(\cdot), \tilde{F}_1(\cdot), \tilde{F}_2(\cdot) : \mathbb{R}^d \to \mathbb{R}^d$ are parts of the augmented surrogate model. By carefully designing the interaction between the augmented parameters and the original parameters, we could turn the transformation to a resolvable one and it does not change the original surrogate model when constraining to the original input $x$ and $\theta$. Specifically, we design the function $\tilde{F}_1$ and $\tilde{F}_2$ as follows:

$$\tilde{F}_1(\tilde{\theta}) = \begin{pmatrix} \theta \\ F_1(\theta) + \theta' \end{pmatrix}, \tilde{F}_2(\tilde{x}) = \begin{pmatrix} x' \\ F_2(x) \end{pmatrix}, \tilde{H}(\tilde{x}) = \begin{bmatrix} I_{d-n} & \\ & H(x) \end{bmatrix}. \tag{9}$$

Before stating our main theorem, we introduce several notations $\tilde{z}_\xi = \tilde{F}_1(\tilde{\xi}) + \tilde{F}_2(\tilde{x})$, $\tilde{z}_\theta = \tilde{F}_1(\tilde{\theta}) + \tilde{F}_2(\tilde{x})$, $\tilde{y}_\xi = (y'_\xi, y_\xi)^T = \tilde{H}(\tilde{F}_1(\tilde{\xi}) + \tilde{F}_2(\tilde{x}))$ and $\tilde{y}_\theta = (y'_\theta, y_\theta)^T = \tilde{H}(\tilde{F}_1(\tilde{\theta}) + \tilde{F}_2(\tilde{x}))$ for simplicity. Then we theoretically prove that the GSmooth classifier is certifiably robust within a given range:

**Theorem 2.** *Suppose $f(x)$ is a classifier and $\tilde{G}(\tilde{x})$ is the smoothed classifier defined in Eq. (7), if there exist $\underline{p_A}$ and $\overline{p_B}$ satisfying*

$$\tilde{G}(\tilde{x})_A \geqslant \underline{p_A} \geqslant \overline{p_B} \geqslant \tilde{G}(\tilde{x})_B,$$

*then $y_A = \arg\max_{i \in \mathcal{Y}} \tilde{G}(\tilde{\tau}(\tilde{\xi}, \tilde{x}))_i$ for any $\|\xi\|_2 \leqslant R$, where*

$$R = \frac{1}{2M^*} \int_{\overline{p_B}}^{\underline{p_A}} \frac{1}{\Phi(p)} dp, \tag{10}$$

*and the coefficient $M^*$ is defined as*

$$M^* = \max_{\xi, \theta \in P} \sqrt{1 + \left\| \frac{\partial F_2(y_\xi)}{\partial \xi} - \frac{\partial F_1(\theta)}{\partial \theta} \right\|_2^2}. \tag{11}$$

As the main result of our GSmooth, we have several observations about it. First, we see that the certified radius is similar to the result in Theorem 1. Second, compared with resolvable transformations, we need to add a new type of noise when constructing the GSmooth classifier. This isotropic noise has the same dimension as the data and is added to the intermediate layers of surrogate neural networks. The theoretical explanation behind this is that this isotropic noise makes the Jacobian matrix of the semantic transformation to be invertible which is crucial for the proof. Third, we observe that the coefficient $M^*$ depends on the norm of the difference of two Jacobian matrices and is independent with the target classifier, later we will discuss the meaning of this term in detail.

Before diving into our theoretical insight and proof of the theorem, we introduce a specific case of Theorem 2 which is more convenient for practical usage. We empirically found that taking a

linear transformation as $F_1(\theta) = A_1\theta + b_1$ where $A_1 \in \mathbb{R}^{n \times m}$, $b_1 \in \mathbb{R}^n$ does not sacrifice the precision of the surrogate network and we have $\frac{\partial F_1(\theta)}{\partial \theta} = A_1$. After substituting the term in Eq. (44) we only need to optimize $\xi$ for calculating $M^*$ and we make the bound tighter. Additionally, we use two gaussian distributions for the noise distribution and the augmented noise distribution, i.e. $g(\theta) = \mathcal{N}(0, \sigma_1^2 I)$ and $g'(\theta') = \mathcal{N}(0, \sigma_2^2 I)$. Formally, we have the following corollary,

**Corollary 1.** *Suppose $f(x)$ is a classifier and $\tilde{G}(\tilde{x})$ is the smoothed classifier defined in Eq. (7), if the layer $F_1(\theta)$ in the surrogate neural network has the following form:*

$$F_1(\theta) = A_1\theta + b_1 \tag{12}$$

*where $A_1 \in \mathbb{R}^{n \times m}$, $b_1 \in \mathbb{R}^n$ are the parameters; and if there exists $\underline{p_A}$ and $\overline{p_B}$ satisfying*

$$\tilde{G}(\tilde{x})_A \geqslant \underline{p_A} \geqslant \overline{p_B} \geqslant \tilde{G}(\tilde{x})_B,$$

*then $y_A = \arg\max_{i \in \mathcal{Y}} \tilde{G}(\tilde{\tau}(\tilde{\xi}, \tilde{x}))_i$ for any $\|\xi\|_2 \leqslant R$ where*

$$R = \frac{1}{2M^*} \left( \Psi\left(\underline{p_A}\right) - \Psi\left(\overline{p_B}\right) \right), \tag{13}$$

*where $\Psi(\cdot)$ is the inverse CDF of standard Gaussian distribution, and the coefficient $M^*$ is defined as,*

$$M^* = \max_{\xi \in P} \sqrt{\frac{1}{\sigma_1^2} + \frac{1}{\sigma_2^2} \left\| \frac{\partial F_2(y_\xi)}{\partial \xi} - A_1 \right\|_2^2}. \tag{14}$$

## 4 PROOF SKETCH AND THEORETICAL ANALYSIS

### 4.1 PROOF SKETCH OF OUR MAIN THEOREM

In this section, we briefly summarize the main idea for proving the Theorem 2 and the theoretical insight of our GSmooth. The key idea is to prove that the gradient of the smoothed classifier can be bounded by a function of the classification confidence and the parameters of the noise distribution. Formally, we calculate the gradient to the perturbation parameter $\xi$ for our augmented smoothed classifier as

$$\nabla_{\tilde{\xi}} \tilde{G}(\tilde{\tau}(\tilde{\xi}, \tilde{x})) = \nabla_{\tilde{\xi}} \mathbb{E}_{\tilde{\theta} \sim \tilde{g}(\tilde{\theta})}[\tilde{f}(\tilde{\tau}(\tilde{\theta}, \tilde{\tau}(\tilde{\xi}, \tilde{x})))]. \tag{15}$$

We expand the expectation into integral and see that

$$\nabla_{\tilde{\xi}} \tilde{G}(\tilde{\tau}(\tilde{\xi}, \tilde{x})) = \int_{\mathbb{R}^{n+d}} \frac{\partial \tilde{f}(\tilde{\tau}(\tilde{\theta}, \tilde{y}_\xi))}{\partial \tilde{\xi}} \tilde{g}(\tilde{\theta}) \mathrm{d}\tilde{\theta}. \tag{16}$$

The key step is to eliminate the gradient of $\frac{\partial \tilde{F}(\tilde{\tau}(\tilde{\theta}, \tilde{y}_\xi))}{\partial \tilde{\xi}}$ and replace it with $\frac{\partial \tilde{f}(\tilde{\tau}(\tilde{\theta}, \tilde{y}_\xi))}{\partial \tilde{\theta}}$. Then we integrate it by parts to get the following obejective,

$$\nabla_{\tilde{\xi}} \tilde{G}(\tilde{\tau}(\tilde{\xi}, \tilde{x})) = - \int_{\mathbb{R}^{n+d}} \tilde{F}(\tilde{\tau}(\tilde{\theta}, \tilde{y}_\xi)) \frac{\partial}{\partial \tilde{\theta}} (\tilde{M}(\tilde{\xi}, \tilde{\theta}) \tilde{g}(\tilde{\theta})) \mathrm{d}\tilde{\theta}. \tag{17}$$

After that, we could bound the gradient of the GSmooth classifier using the technique similar to randomized smoothing (Yang et al., 2020). More details of the proof can be found in Appendix A.

### 4.2 THEORETICAL INSIGHT

Next, we provide the theoretical insight for our augmentation scheme on transformation parameters and data. First, the key is to expand the transformation space by adding additional dimensions to form a closed space. In the augmented space, the Jacobian matrix of the semantic transformation became invertible which is crucial for our proof. Second, as we can see in Eq. (14) that $M^*$ is influenced by two factors. One is the standard deviation of two noise distributions. The other is the norm of the Jacobian matrix $\frac{\partial F_2(y_\xi)}{\partial \xi} - A_1$. It can be viewed as the residual of the non-resolvable part of the transformation. Along this line, our method decomposes the unknown semantic transformation into a resolvable part and a residual part. The non-resolvable residual part could be handled by introducing additional noise with standard deviation $\sigma_2$.

### 4.3 ERROR ANALYSIS FOR SURROGATE MODEL APPROXIMATION

In this subsection, we theoretically analyze the effectiveness of certifying real semantic transformation due to the existence of approximation error of surrogate neural networks.

**Theorem 3.** *Suppose the simulation of the semantic transformation has an small enough error*

$$\|\tilde{\tau}(\tilde{\xi}, \tilde{x}) - \overline{\tau}(\tilde{\xi}, \tilde{x})\|_2 < \varepsilon,$$

*Then there exists a constant ratio $A = A(\|F_1'(\tilde{\xi})\|_2, \|F_2'(\tilde{y}_\xi)\|_2, \|F_2'(\tilde{z}_\xi)\|_2) > 0$ does not depend on the target classifier, we have the certified radius for the real semantic transformation satisfies that*

$$R_r > R(1 - A\varepsilon)$$

*where $R$ is the certified radius for surrogate the neural network in Theorem 2 and*

$$R = \frac{1}{2M^*} \int_{\overline{p_B}}^{p_A} \frac{1}{\Phi(p)} dp,$$

We find that the reduction of the certified radius is influenced by two factors. The first one is the approximation error $\epsilon$ between the surrogate transformation and the real semantic transformation. The second one the ratio $A$ is about the norm of the Jacobian matrix for some layers of the surrogate model which is also an inherent property of the semantic transformation itself and does not depend on the target classifier.

## 5 EXPERIMENTS

### 5.1 EXPERIMENTAL SETUP AND EVALUATION METRICS

In this section, we conduct extensive experiments to show the effectiveness of our GSmooth on various types of semantic transformations. We use MNIST, CIFAR-10 and CIFAR-100 (Krizhevsky et al., 2009) datasets to verify our methods. We train a resnet with 110 layers (He et al., 2016a) from scratch. Similar to prior works, we apply moderate data augmentation (Cohen et al., 2019) to improve the generalization of the classifier. For the surrogate image-to-image translation model for simulating semantic transformations, we adopt the U-Net architecture (Ronneberger et al., 2015) for $\tilde{H}(\cdot)$ layers and several simple convolutional or linear layers for $\tilde{F}_1(\cdot)$ and $\tilde{F}_2(\cdot)$. All models are trained using Adam optimizer with an initial learning rate of 0.001 that decays every 50 epochs until convergence. The algorithms for calculating $M^*$ and other details in our experiments are listed in Appendix B due to limited space. The evaluation metric is the certified accuracy, which is the percentage of samples that are correctly classified and has a larger certified radius than the given range. We use $\|\alpha\|$ to indicate the preset certified radius.

### 5.2 MAIN RESULTS

To demonstrate the effectiveness of our GSmooth on certifying complex semantic transformations, we measure the certified correct accuracy for different semantic transformations on different datasets in Table 1. We compare the results of our GSmooth with several baselines, including *randomized smoothing* for some simple semantic transformation of **TSS** (Li et al., 2021) and **IndivSPT/distSPT** (Fischer et al., 2020), and our GSmooth is a natural and powerful extension of their methods. We also compare our method with the *deterministic certification* approaches, including **DeepG** (Balunović et al., 2019) that uses linear relaxations similar to Wong & Kolter (2018), **Interval** (Singh et al., 2019) that is based on interval bound propagation, **VeriVis** (Pei et al., 2017) that enumerates all possible outcomes for semantic transformations with finite values of parameters, and **Semanify-NN** (Mohapatra et al., 2020) which uses a new preprocessing layer to turn the problem into a $\ell_p$ norm certification.

We have the following observations for the experimental results. First, only our method achieves non-zero accuracy on certifying some complex semantic transformations and the results verify our Theorem 2. This is a breakthrough that greatly extends the boundary of randomized smoothing based methods. Second, we see the performance of GSmooth is similar to the state-of-the-art randomized smoothing approaches like TSS on several simple semantic transformations like Gaussian

| Cert Acc(%) | Type | Dataset | Attack Range | Certified Accuracy(%) | | | | | | |
|---|---|---|---|---|---|---|---|---|---|---|
| | | | | GSmooth (Ours) | TSS | DeepG | Interval | VeriVis | Semanify-NN | IndivSPT/distSPT |
| Gaussian Blur | Additive | MNIST | $\|\alpha\|_2 < 6$ | **91.0** | 90.6 | – | – | – | – | – |
| | | CIFAR-10 | $\|\alpha\|_2 < 4$ | **67.4** | 63.6 | – | – | – | – | – |
| | | CIFAR-100 | | **22.1** | 21.0 | – | – | – | – | – |
| Translation | Additive | MNIST | $\|\alpha\|_2 < 8$ | 98.7 | **99.6** | 0.0 | 0.0 | 98.8 | 98.8 | 99.6 |
| | | CIFAR-10 | $\|\alpha\|_2 < 20$ | **82.2** | 80.8 | 0.0 | 0.0 | 65.0 | 65.0 | 78.8 |
| | | CIFAR-100 | | **42.2** | 41.3 | – | – | 24.2 | 24.2 | – |
| Brightness, Contrast | Resolvable | MNIST | $\|\alpha\|_\infty < 0.5$ | **97.7** | 97.6 | ⩽0.4 | 0.0 | – | ⩽74 | – |
| | | CIFAR-10 | $\|\alpha\|_\infty < 0.4$ | **82.5** | 82.4 | 0.0 | 0.0 | – | – | – |
| | | CIFAR-100 | | **42.3** | 41.4 | 0.0 | 0.0 | – | – | – |
| Rotation | Non-resolvable | MNIST | $\|\alpha\|_2 < 50°$ | 95.7 | **97.4** | ⩽85.8 | ⩽6.0 | – | ⩽92.48 | ⩽76 |
| | | CIFAR-10 | $\|\alpha\|_2 < 10°$ | 64.6 | **70.6** | 62.5 | 20.2 | – | ⩽49.37 | ⩽34 |
| | | CIFAR-100 | | 33.2 | **36.7** | 0.0 | 0.0 | – | ⩽21.7 | ⩽18 |
| Scaling | Non-resolvable | MNIST | $\|\alpha\|_2 < 0.3$ | 95.9 | **97.2** | 85.0 | 16.4 | – | – | – |
| | | CIFAR-10 | $\|\alpha\|_2 < 0.3$ | 54.3 | **58.8** | 0.0 | 0.0 | – | – | – |
| | | CIFAR-100 | | 31.2 | **37.8** | 0.0 | 0.0 | – | – | – |
| Rotational Blur | Non-resolvable | MNIST | $\|\alpha\|_2 < 10$ | **95.9** | – | – | – | – | – | – |
| | | CIFAR-10 | $\|\alpha\|_2 < 10$ | **39.7** | – | – | – | – | – | – |
| | | CIFAR-100 | | **27.2** | – | – | – | – | – | – |
| Defocus Blur | Non-resolvable | MNIST | $\|\alpha\|_2 < 5$ | **89.2** | – | – | – | – | – | – |
| | | CIFAR-10 | $\|\alpha\|_2 < 5$ | **25.0** | – | – | – | – | – | – |
| | | CIFAR-100 | | **13.1** | – | – | – | – | – | – |
| Zoom Blur | Non-resolvable | MNIST | $\|\alpha\|_2 < 0.5$ | **93.9** | – | – | – | – | – | – |
| | | CIFAR-10 | $\|\alpha\|_2 < 0.5$ | **44.6** | – | – | – | – | – | – |
| | | CIFAR-100 | | **14.2** | – | – | – | – | – | – |
| Pixelate | Non-resolvable | MNIST | $\|\alpha\|_2 < 0.5$ | **87.1** | – | – | – | – | – | – |
| | | CIFAR-10 | $\|\alpha\|_2 < 0.5$ | **45.3** | – | – | – | – | – | – |
| | | CIFAR-100 | | **30.2** | – | – | – | – | – | – |

Table 1: Our main results of certification accuracy on several datasets and multiple types of semantic transformations. – or 0.0% means the method fails to certify this type of semantic transformation.

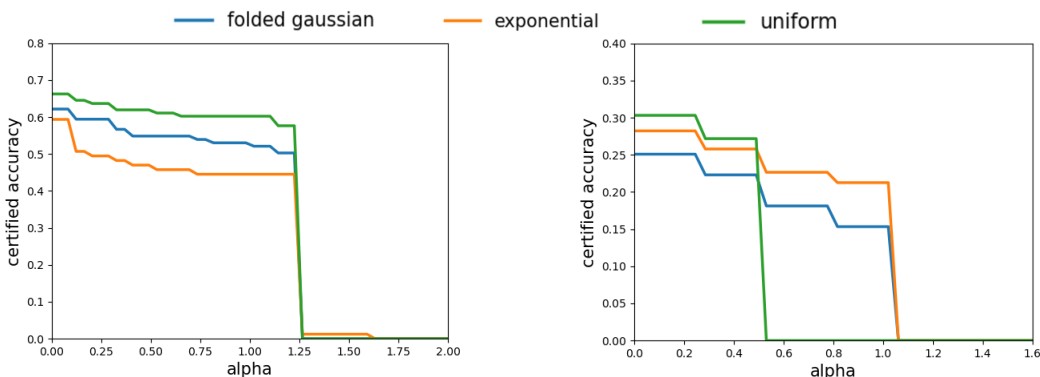

Figure 2: Results of ablation experiments on the influence of smoothing distribution for zoomed blur on CIFAR-10 dataset. The horizontal axis $\|\alpha\|_2$ is the certified raidus.

blur, translation. This is a natural result since our method works similarly for resolvable transformations. For two specific non-resolvable transformations, i.e. rotation and scaling, our accuracy is slightly lower. The possible reason is that in TSS (Li et al., 2021) they derive more elaborate Lipschitz bound for rotation which is better than us. Third, those inherently non-resolvable transformations like image blurring (except Gaussian blur) and pixelate are more difficult than resolvable or approximately resolvable (rotation) transformations. Thus their certified accuracy is also lower.

## 5.3 ABLATION STUDY

**Ablation study on the influence of noise distribution.** The choice of noise distribution is important for randomized smoothing based methods. Since our GSmooth could certify different types of semantic transformations that exhibit different properties. Understanding the influence of different smoothing distributions for different semantic transformations is necessary. We choose (folded) Gaussian, uniform, and exponential distribution and compare the certified accuracy on zoom blur transformation for both CIFAR-10 and CIFAR-100 datasets. As shown in Fig. 2, We found that

| Cert Acc | CIFAR-10 | | | | | CIFAR-100 | | | | |
|---|---|---|---|---|---|---|---|---|---|---|
| | $\sigma_1$ \\ $\sigma_2$ | 0.05 | 0.10 | 0.15 | 0.25 | $\sigma_1$ \\ $\sigma_2$ | 0.05 | 0.10 | 0.15 | 0.20 |
| Rotational Blur | 0.1 | 44.3 | 46.4 | 35.5 | 16.1 | 0.1 | 23.1 | 26.2 | 17.2 | 10.4 |
| | 0.25 | 45.7 | 47.1 | 38.3 | 18.9 | 0.25 | 23.2 | **27.2** | 20.3 | 11.3 |
| | 0.5 | 46.5 | **48.4** | 38.8 | 18.3 | 0.5 | 22.2 | 26.1 | 18.6 | 11.8 |
| | 0.75 | 42.1 | 45.5 | 36.6 | 17.0 | 0.75 | 24.0 | 25.5 | 18.3 | 13.2 |

Table 2: Results of ablation study on the influence of standard deviation of smoothing distributions, i.e. transformation noise $\sigma_1$ and augmented noise $\sigma_2$ for certification accuracy on rotational blur.

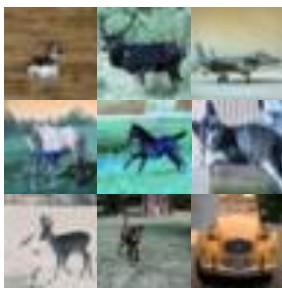 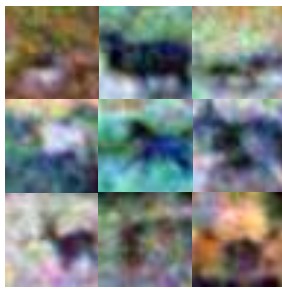 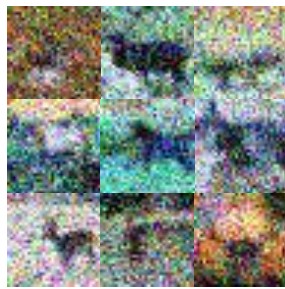

Figure 3: Visualization of difference between the augmented noise in the semantic layers and the noise on raw images. Left: original images from CIFAR-10. Middle: images with augmented noise of $\sigma_2 = 0.2$. Right: images with additive Gaussian noise $\sigma = 0.2$.

the impact of smoothing distributions depends on datasets. Uniform distribution is better for small radius certification while exponential distribution is more suitable for certifying large radius on average.

**Ablation study on the influence of noise variance for certification.** Since our GSmooth contains two different variances for controlling the resolvable part and the residual part for a non-resolvable semantic transformation. Here we investigate the effect of different noise variance on the certified accuracy. The results are shown in Table 2. We found that using medium transformation noise and augmented noise achieves the best certified accuracy. The fact is consistent with results in (Cohen et al., 2019). An explanation is that there is a trade-off since higher the noise variance decreases the coefficient $M^*$ but it might also degrade the clean accuracy.

**Visulization experiments: comparsion between augmented semantic noise and image noise.** Our GSmooth needs to add a new noise in the semantic layers of the surrogate model. Here we compare the difference between these two types of noise and visualize them. We random sample images from CIFAR-10 and add gaussian noise with $\sigma = 0.2$ to both the semantic layer of the surrogate model simulating zoomed blur transformation and raw images. Results are shown in Fig. 3. Both two types of noise severely blur the images. But we found that the augmented semantic noise is more placid which can therefore keep the holistic semantic features better, e.g., shapes.

## 6 CONCLUSIONS

In this paper, we proposed a generalized randomized smoothing framework (GSmooth) for certifying robustness against general semantic transformations. We proposed a novel idea that using a surrogate neural network to fit semantic transformations. Then we prove tight certified robustness bound for the surrogate model and use it for certifying semantic transformations. Extensive experiments verify the effectiveness of our method and we achieved state-of-the-art performance on various types of semantic transformations. In the future, we plan to extend our method to real-world scenarios on more diverse semantic transformations.

## 7 REPRODUCIBILITY STATEMENT

We ensure the reproducibility of our paper from three aspects. (1) Experiment: The implementation of our experiment is described in Sec. 5.1. Ablation study for our experiments is in Sec. 5.3. Further details are in Appendix B. (2) Code: Our code is included in supplementary materials. (3) Theory and Method: A complete proof of the theoretical results described is provided in Appendix A.

## 8 ETHICS STATEMENT

Machine learning models are easily attacked by adversarial examples and semantic transformations. Thus it is fundamental problem to develop certified robust machine learning methods. This paper proposed GSmooth to certify against semantic transformations. It may promote the development of safe and reliable machine learning models in the future.

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

## A    PROOF OF THEOREMS

In this section, we will provide detailed proofs of theorems in our paper.

First, we restate the theorem of randomized smoothing for additive noise and binary classifiers $f(\cdot) : \mathbb{R}^n \to [0, 1]$,

$$G(x) = \mathbb{E}_{\theta \sim g}[f(x + \theta)] \tag{18}$$

**Theorem 4.** *Let $f(x)$ be any classifier and $G(x)$ be the smoothed classifier defined in Eq. (18), if $G(x) < \frac{1}{2}$, then $G(x + \delta) < \frac{1}{2}$ for any*

$$\|\delta\| < \int_{G(x)}^{\frac{1}{2}} \frac{1}{\Phi(p)} \mathrm{d}p \tag{19}$$

*where $\Phi(\cdot)$ is a function about smoothing distribution defined in Eq. (2).*

*Proof.* We first calculate the gradient of the smoothed classifier

$$
\begin{aligned}
\nabla G(x) &= \frac{\partial}{\partial x} \int_{\mathbb{R}^n} f(x + \theta) g(\theta) \mathrm{d}\theta \\
&= \int_{\mathbb{R}^n} \frac{\partial}{\partial x} f(x + \theta) g(\theta) \mathrm{d}\theta \\
&= \int_{\mathbb{R}^n} \frac{\partial}{\partial \theta} f(x + \theta) g(\theta) \mathrm{d}\theta
\end{aligned}
$$

Then we multiple any vector with unit norm $u \in \mathcal{B}_n(1) = \{u : \|u\| = 1, u \in \mathbb{R}^n\}$,

$$
\begin{aligned}
|\langle \nabla G(x), u \rangle| &= \left| \left\langle \int_{\mathbb{R}^n} \frac{\partial}{\partial \theta} f(x + \theta) g(\theta) \mathrm{d}\theta, u \right\rangle \right| \\
&= \left| \int_{\mathbb{R}^n} \left\langle \frac{\partial}{\partial \theta} f(x + \theta), u \right\rangle g(\theta) \mathrm{d}\theta \right| \\
&= \left| \sum_i \int_{\mathbb{R}^n} \frac{\partial f(x + \theta)}{\partial \theta_i} u_i g(\theta) \mathrm{d}\theta \right| \\
&= \left| \sum_i \int_{\mathbb{R}^{n-1}} \left( \int_{-\infty}^{+\infty} \frac{\partial f(x + \theta)}{\partial \theta_i} u_i g(\theta) \mathrm{d}\theta_i \right) \prod_{j \neq i} \mathrm{d}\theta_j \right| \\
&= \left| -\sum_i \int_{\mathbb{R}^{n-1}} \left( \int_{-\infty}^{+\infty} f(x + \theta) u_i \frac{\partial g(\theta)}{\partial \theta_i} \mathrm{d}\theta_i \right) \prod_{j \neq i} \mathrm{d}\theta_j \right| \\
&= \left| \sum_i \int_{\mathbb{R}^n} f(x + \theta) \frac{\partial g(\theta)}{\partial \theta_i} u_i \mathrm{d}\theta \right| \\
&= \left| \int_{\mathbb{R}^n} f(x + \theta) \langle \nabla g(\theta), u \rangle \mathrm{d}\theta \right| \\
&= \left| \int_{\mathbb{R}^n} f(x + \theta) g(\theta) \langle \nabla \psi(\theta), u \rangle \mathrm{d}\theta \right| \\
&= |\mathbb{E}_{\theta \sim g}[f(x + \theta) \langle \nabla \psi(\theta), u \rangle]| \tag{20}
\end{aligned}
$$

To bound the gradient of the smoothed classifier, we use the following inequality,

$$|\langle \nabla G(x), u \rangle| \leqslant \sup_{\hat{f} : \hat{G}(x) = G(x)} \mathbb{E}_{\theta \sim g} \left[ \hat{f}(x + \theta) \langle \psi(\theta), u \rangle \right] \tag{21}$$

As shown in Yang et al. (2020), the optimal $\hat{f}(x)$ achieves at,

$$\hat{f}(x + \theta) = \begin{cases} 1, \text{if } \langle u, \psi(\theta) \rangle > \varphi_u^{-1}(G(x)) \\ 0, \text{else} \end{cases} \tag{22}$$

Then,

$$
\begin{aligned}
\mathbb{E}_{\theta \sim g}\left[\hat{f}(x+\theta)\langle u, \psi(\theta)\rangle\right] &= \mathbb{E}\left[\gamma_u \mathbb{I}\{\gamma_u > \varphi_u^{-1}(G(x))\}\right] \\
&\leqslant \Phi(G(x))
\end{aligned} \tag{23}
$$

which means that,

$$
|\langle \nabla G(x), u\rangle| \leqslant \Phi(G(x)) \tag{24}
$$

and this is true for all $u \in \mathcal{B}_n(1)$, so we have,

$$
\max_{u \in \mathcal{B}_n(1)} \langle \nabla G(x), u\rangle \leqslant \Phi(G(x)) \tag{25}
$$

Consider a path from $\xi_t : [0, \|\delta\|] \to \mathbb{R}^d$ with $\xi_0 = x$ and $\xi_{\|\delta\|} = x + \delta$ and $\xi_t' = \frac{\delta}{\|\delta\|}$, we have

$$
\frac{\mathrm{d}G(\xi_t)}{\mathrm{d}t} = \langle \nabla G(\xi_t), u\rangle \leqslant \Phi(G(\xi_t)). \tag{26}
$$

If the norm of $\delta$ satisfies that,

$$
\|\delta\| < \int_{G(x)}^{\frac{1}{2}} \frac{1}{\Phi(p)} \mathrm{d}p, \tag{27}
$$

if right hand side exists, we name it $\|\delta_0\| = \int_{G(x)}^{\frac{1}{2}} 1/\Phi(p)\mathrm{d}p$. Without loss of generality, we assume that $G(\xi_t)$ is increasing in $t$, then we could calculate the minimal $t$ when $G(\xi_t)$ increase to $\frac{1}{2}$,

$$
T = \int_{G(\xi_0)}^{\frac{1}{2}} \frac{1}{\Phi(p)} \mathrm{d}p = \|\delta_0\|. \tag{28}
$$

By the generality of $\delta$ we have,

$$
G(x + \delta) < \frac{1}{2} \tag{29}
$$

for any $\delta < \|\delta_0\|$. $\qquad \square$

This theorem is naturally extended to problems with $p > 2$ classes by considering the two top classes which are $G(x)_A$ and $G(x)_B$. This turns the problem into a binary classification problem.

## A.1 THE PROOF OF THEOREM 1

**Theorem 1.** *Let $f(x)$ be any classifier and $G(x)$ be the smoothed classifier defined in Eq. (1), if there exists a function $M(\cdot)$, and the transformation $\tau(\cdot, \cdot)$ satisfies that*

$$
\frac{\partial \gamma(\theta, \xi)}{\partial \xi} = \frac{\partial \gamma(\theta, \xi)}{\partial \theta} M(\theta, \xi),
$$

*and there exists two constants $\underline{p_A}$, $\overline{p_B}$ that satisfies that*

$$
G(x)_A \geqslant \underline{p_A} \geqslant \overline{p_B} \geqslant G(x)_B,
$$

*then $y_A = \arg\max_{i \in \mathcal{Y}} G(\tau(\xi, x))_i$ holds for any $\|\xi\| \leqslant R$ where*

$$
R = \frac{1}{2M^*} \int_{\overline{p_B}}^{\underline{p_A}} \frac{1}{\Phi(p)} dp, \tag{30}
$$

*here $M^* = max_{\xi, \theta}\|M(\xi, \theta)\|$.*

*Proof.* WLOG, we only prove it for binary cases that $f(\cdot) : \mathbb{R}^n \to [0, 1]$.

$$
\begin{aligned}
&\nabla_\xi G(\tau(\gamma(\theta, \xi), x)) \\
&= \int \frac{\partial f(\tau(\gamma(\theta, \xi), x))}{\partial \tau(\gamma(\theta, \xi), x)} \cdot \frac{\partial \tau(\gamma(\theta, \xi), x)}{\partial \xi} g(\theta) \mathrm{d}\theta \\
&= \int \frac{\partial f(\tau(\gamma(\theta, \xi), x))}{\partial \tau(\gamma(\theta, \xi), x)} \cdot \frac{\partial \tau(\gamma(\theta, \xi), x)}{\partial \gamma(\theta, \xi)} \frac{\partial \gamma(\theta, \xi)}{\partial \xi} g(\theta) \mathrm{d}\theta \\
&= \int \frac{\partial f(\tau(\gamma(\theta, \xi), x))}{\partial \tau(\gamma(\theta, \xi), x)} \cdot \frac{\partial \tau(\gamma(\theta, \xi), x)}{\partial \gamma(\theta, \xi)} \frac{\partial \gamma(\theta, \xi)}{\partial \theta} M(\theta, \xi) g(\theta) \mathrm{d}\theta \\
&= \int \frac{\partial f(\tau(\gamma(\theta, \xi), x))}{\partial \theta} M(\theta, \xi) g(\theta) \mathrm{d}\theta
\end{aligned}
\tag{31}
$$

For $u \in \mathbb{R}^d$ and $\|u\| = 1$, we have:

$$
\begin{aligned}
|\langle \nabla_\xi G(\gamma(\xi, x)), u \rangle| &= \left| \int \frac{\partial f(\tau(\gamma(\theta, \xi), x))}{\partial \theta} M(\theta, \xi) u g(\theta) \mathrm{d}\theta \right| \tag{32} \\
&\leqslant M^* \max_{\|v\|=1} \left| \int \frac{\partial f(\tau(\gamma(\theta, \xi), x))}{\partial \theta} v g(\theta) \mathrm{d}\theta \right| \tag{33} \\
&\leqslant M^* \max_{\|v\|=1} \left| \int f(\tau(\gamma(\theta, \xi), x)) \frac{\partial g(\theta)}{\partial \theta} v \mathrm{d}\theta \right| \tag{34} \\
& \tag{35}
\end{aligned}
$$

here $M^* = \max_{\theta, \xi} \|M(\theta, \xi)\|$ and we assume that $g(\theta) = \exp(-\psi(\theta))$:

$$
\begin{aligned}
|\langle \nabla_\xi G(\gamma(\xi, x)), u \rangle| &\leqslant M^* \max_{\|v\|=1} \left| \int f(\tau(\gamma(\theta, \xi), x)) \frac{\partial g(\theta)}{\partial \theta} v \mathrm{d}\theta \right| \tag{36} \\
&= M^* \max_{\|v\|=1} \left| \int f(\tau(\gamma(\theta, \xi), x)) g(\theta) \nabla \psi(\theta) v \mathrm{d}\theta \right| \tag{37} \\
&= M^* \max_{\|v\|=1} |\mathbb{E}_{\theta \sim g} [f(\tau(\gamma(\theta, \xi), x)) \langle \nabla \psi(\theta), v \rangle]| \tag{38} \\
&\leqslant M^* \max_{\|v\|=1} \sup_{\hat{f}: \hat{G}(\tau(\xi, x)) = G(\tau(\xi, x))} \mathbb{E}_{\theta \sim g} [\hat{f}(\tau(\gamma(\theta, \xi), x)) \langle \psi(\theta), u \rangle] \tag{39}
\end{aligned}
$$

Similar with Theorem 4, the optimal $\hat{f}$ achieves at

$$
\hat{f}(\tau(\gamma(\theta, \xi), x)) = \begin{cases} 1, \text{if} \langle \psi(\theta), u \rangle > \varphi_u^{-1}(G(\tau(\xi, x)))) \\ 0, \text{else} \end{cases}
\tag{40}
$$

Then we have

$$
|\langle \nabla_\xi G(\tau(\xi, x)), u \rangle| \leqslant \Phi(G(\tau(\xi, x))).
\tag{41}
$$

Consider a path from $\zeta_t : [0, \|\delta\|] \to \mathbb{R}^d$ with $\zeta_0 = x$ and $\zeta_{\|\delta\|} = \tau(\xi, x)$ and $\zeta_t' = \frac{\delta}{\|\delta\|}$, we have

$$
\frac{\mathrm{d}G(\xi_t)}{\mathrm{d}t} = \langle \nabla G(\xi_t), u \rangle \leqslant \Phi(G(\xi_t)).
\tag{42}
$$

The last part of proof is the same with Theorem 4. $\qquad \square$

## A.2 THE PROOF OF THEOREM 2

**Theorem 2.** *Suppose $f(x)$ is any classifier and $\tilde{G}(\tilde{x})$ is the smoothed classifier defined in Eq. (7), if there exists $\underline{p_A}$, $\overline{p_B}$ that satisfies that*

$$
\tilde{G}(\tilde{x})_A \geqslant \underline{p_A} \geqslant \overline{p_B} \geqslant \tilde{G}(\tilde{x})_B,
$$

*then $y_A = \arg\max_{i\in\mathcal{Y}} \tilde{G}(\tilde{\tau}(\tilde{\xi},\tilde{x}))_i$ for any $\|\xi\|_2 \leqslant R$ where*

$$R = \frac{1}{2M^*}\int_{\overline{p_B}}^{p_A} \frac{1}{\Phi(p)}dp, \tag{43}$$

*and the coefficient $M^*$ is defined as*

$$M^* = \max_{\xi,\theta\in P}\sqrt{1 + \left\|\frac{\partial F_2(y_\xi)}{\partial \xi} - \frac{\partial F_1(\theta)}{\partial \theta}\right\|_2^2}. \tag{44}$$

*Proof.* In this part, we will prove Theorem 2, which is the main result in this paper. WLOG, we prove it for binary cases where $f(\cdot): \mathbb{R}^n \to [0,1]$. First, we will calculate the gradient of $\tilde{G}(\tilde{\tau}(\tilde{\xi},\tilde{x}))$ to $\tilde{\xi}$:

$$\nabla_{\tilde{\xi}}\tilde{G}(\tilde{\tau}(\tilde{\xi},\tilde{x})) = \nabla_{\tilde{\xi}}\mathbb{E}_{\tilde{\theta}\sim\tilde{g}(\tilde{\theta})}[\tilde{f}(\tilde{\tau}(\tilde{\theta},\tilde{\tau}(\tilde{\xi},\tilde{x})))]. \tag{45}$$

We expand the expectation into integral and use chain rule to see that

$$\nabla_{\tilde{\xi}}\tilde{G}(\tilde{\tau}(\tilde{\xi},\tilde{x})) = \int_{\mathbb{R}^{n+d}} \frac{\partial \tilde{f}(\tilde{\tau}(\tilde{\theta},\tilde{y}_\xi))}{\partial \tilde{\tau}(\tilde{\theta},\tilde{y}_\xi)} \cdot \frac{\partial \tilde{\tau}(\tilde{\theta},\tilde{y}_\xi)}{\partial \tilde{y}_\xi} \cdot \frac{\partial \tilde{y}_\xi}{\partial \tilde{\xi}}\tilde{g}(\tilde{\theta})\mathrm{d}\tilde{\theta}. \tag{46}$$

The key step is to eliminate the gradient of $\frac{\partial \tilde{f}(\tilde{\tau}(\tilde{\theta},\tilde{y}_\xi))}{\partial \tilde{\tau}(\tilde{\theta},\tilde{y}_\xi)}$ and replace it with $\frac{\partial \tilde{f}(\tilde{\tau}(\tilde{\theta},\tilde{y}_\xi))}{\partial \tilde{\theta}}$. Since:

$$\frac{\partial \tilde{\tau}(\tilde{\theta},\tilde{y}_\xi)}{\partial \tilde{y}_\xi} = \begin{bmatrix} \sigma_1'(z_\theta') & \\ & \sigma_2'(z_\theta) \end{bmatrix}\begin{bmatrix} F_{21}'(y_\xi') & \\ & F_{22}'(y_\xi) \end{bmatrix}, \tag{47}$$

$$\frac{\partial \tilde{y}_\xi}{\partial \tilde{\xi}} = \begin{bmatrix} H_1'(z_\xi') & \\ & H_2'(z_\xi) \end{bmatrix}\begin{bmatrix} I_d & \\ F_1'(\xi) & I_n \end{bmatrix}. \tag{48}$$

We have:

$$\begin{aligned}
\nabla_{\tilde{\xi}}\tilde{G}(\tilde{\tau}(\tilde{\xi},\tilde{x})) &= \int_{\mathbb{R}^{n+d}} \frac{\partial \tilde{F}(\tilde{\tau}(\tilde{\theta},\tilde{y}_\xi))}{\partial \tilde{\tau}(\tilde{\theta},\tilde{y}_\xi)} \cdot \begin{bmatrix} H_1'(z_\theta') & \\ & H_2'(z_\theta) \end{bmatrix}\begin{bmatrix} F_{21}'(y_\xi') & \\ & F_{22}'(y_\xi) \end{bmatrix} \\
&\qquad \begin{bmatrix} H_1'(z_\xi') & \\ & H_2'(z_\xi) \end{bmatrix}\begin{bmatrix} I_d & \\ F_1'(\xi) & I_n \end{bmatrix}\tilde{g}(\tilde{\theta})\mathrm{d}\tilde{\theta} \\[6pt]
&= \int_{\mathbb{R}^{n+d}} \frac{\partial \tilde{F}(\tilde{\tau}(\tilde{\theta},\tilde{y}_\xi))}{\partial \tilde{\tau}(\tilde{\theta},\tilde{y}_\xi)} \cdot \begin{bmatrix} H_1'(z_\theta') & \\ & H_2'(z_\theta) \end{bmatrix}\begin{bmatrix} I_d & \\ F_1'(\theta) & I_n \end{bmatrix}\begin{bmatrix} I_d & \\ -F_1'(\theta) & I_n \end{bmatrix} \\
&\qquad \begin{bmatrix} F_{21}'(y_\xi') & \\ & F_{22}'(y_\xi) \end{bmatrix}\begin{bmatrix} H_1'(z_\xi') & \\ & H_2'(z_\xi) \end{bmatrix}\begin{bmatrix} I_d & \\ F_1'(\xi) & I_n \end{bmatrix}\tilde{g}(\tilde{\theta})\mathrm{d}\tilde{\theta} \\[6pt]
&= \int_{\mathbb{R}^{n+d}} \frac{\partial \tilde{F}(\tilde{\tau}(\tilde{\theta},\tilde{y}_\xi))}{\partial \tilde{\theta}} \begin{bmatrix} I_d & \\ -F_1'(\theta) & I_n \end{bmatrix}\begin{bmatrix} F_{21}'(y_\xi') & \\ & F_{22}'(y_\xi) \end{bmatrix} \\
&\qquad \begin{bmatrix} H_1'(z_\xi') & \\ & H_2'(z_\xi) \end{bmatrix}\begin{bmatrix} I_d & \\ F_1'(\xi) & I_n \end{bmatrix}\tilde{g}(\tilde{\theta})\mathrm{d}\tilde{\theta} \\[6pt]
&= \int_{\mathbb{R}^{n+d}} \frac{\partial \tilde{F}(\tilde{\tau}(\tilde{\theta},\tilde{y}_\xi))}{\partial \tilde{\theta}}\tilde{M}(\tilde{\xi},\tilde{\theta})g(\tilde{\theta})\mathrm{d}\tilde{\theta},
\end{aligned}$$

$$\begin{aligned}
&\tag{49}\\
&\tag{50}\\
&\tag{51}\\
&\tag{52}
\end{aligned}$$

here we define

$$\tilde{M}(\tilde{\xi},\tilde{\theta}) \triangleq \begin{bmatrix} I_d & \\ -F_1'(\theta) & I_n \end{bmatrix}\begin{bmatrix} F_{21}'(y_\xi') & \\ & F_{22}'(y_\xi) \end{bmatrix}\begin{bmatrix} H_1'(z_\xi') & \\ & H_2'(z_\xi) \end{bmatrix}\begin{bmatrix} I_d & \\ F_1'(\xi) & I_n \end{bmatrix} \tag{53}$$

We consider the Unit enlargement, which means:

$$H_1(z') = z', F_{21}(x') = x' \tag{54}$$

thus:

$$\tilde{M}(\tilde{\xi}, \tilde{\theta}) = \begin{bmatrix} I_d & O_{d \times n} \\ F_{21}'(y_\xi) - F_1'(\theta) & F_{22}'(y_\xi) H_2'(z_\xi) \end{bmatrix} \tag{55}$$

Since $\theta'$ is the virtual parameter introduced, which can be taken as 0 in case of actual disturbance. Thus we only need to consider the projection of $\nabla_{\tilde{\xi}} \tilde{G}(\tilde{y}_\xi)$ in the space of $\xi$. Thus we set

$$\tilde{u} = \begin{pmatrix} u \\ O_{n \times 1} \end{pmatrix}, \tag{56}$$

here $u \in \mathbb{R}^d$ and $\|u\| = 1$. Assume

$$\tilde{g}(\tilde{\theta}) = \exp(-\tilde{\psi}(\tilde{\theta})) \tag{57}$$

$$\frac{\partial \tilde{g}(\tilde{\theta})}{\partial \tilde{\theta}} = -\tilde{g}(\tilde{\theta}) \cdot \nabla \tilde{\psi}(\tilde{\theta}). \tag{58}$$

And we have

$$\left| \langle \nabla_{\tilde{\xi}} \tilde{G}(\tilde{y}_\xi), \tilde{u} \rangle \right| = \left| \int_{\mathbb{R}^{n+d}} \frac{\partial \tilde{f}(\tilde{\tau}(\tilde{\theta}, \tilde{y}_\xi))}{\partial \tilde{\theta}} \tilde{M}(\tilde{\xi}, \tilde{\theta}) \tilde{u} \tilde{g}(\tilde{\theta}) \mathrm{d}\tilde{\theta} \right| \tag{59}$$

$$= \left| \int_{\mathbb{R}^{n+d}} \frac{\partial \tilde{f}(\tilde{\tau}(\tilde{\theta}, \tilde{y}_\xi))}{\partial \tilde{\theta}} \begin{bmatrix} I_d, & O_{d \times n} \\ F_{22}'(y_\xi) - F_1'(\theta), & O_{n \times n} \end{bmatrix} \tilde{u} \tilde{g}(\tilde{\theta}) \mathrm{d}\tilde{\theta} \right| \tag{60}$$

$$= \left| \int_{\mathbb{R}^{n+d}} \frac{\partial \tilde{f}(\tilde{\tau}(\tilde{\theta}, \tilde{y}_\xi))}{\partial \tilde{\theta}} M(\xi, \theta) \tilde{u} \tilde{g}(\tilde{\theta}) \mathrm{d}\tilde{\theta} \right| \tag{61}$$

$$\leqslant M^* \max_{\|\tilde{v}\|_2 = 1} \left| \int_{\mathbb{R}^{n+d}} \frac{\partial \tilde{f}(\tilde{\tau}(\tilde{\theta}, \tilde{y}_\xi))}{\partial \tilde{\theta}} \tilde{v} \tilde{g}(\tilde{\theta}) \mathrm{d}\tilde{\theta} \right| \tag{62}$$

$$= M^* \max_{\|\tilde{v}\|_2 = 1} \left| \int_{\mathbb{R}^{n+d}} \tilde{f}(\tilde{\tau}(\tilde{\theta}, \tilde{y}_\xi)) \frac{\partial \tilde{g}(\tilde{\theta})}{\partial \tilde{\theta}} \tilde{v} \mathrm{d}\tilde{\theta} \right| \tag{63}$$

$$= M^* \max_{\|\tilde{v}\|_2 = 1} \left| \int_{\mathbb{R}^{n+d}} \tilde{f}(\tilde{\tau}(\tilde{\theta}, \tilde{y}_\xi)) \tilde{g}(\tilde{\theta}) \nabla \tilde{\psi}(\tilde{\theta}) \tilde{v} \mathrm{d}\tilde{\theta} \right| \tag{64}$$

$$= M^* \max_{\|\tilde{v}\|_2 = 1} \left| \mathbb{E}_{\theta \sim g} \left[ \tilde{f}(\tilde{\tau}(\tilde{\theta}, \tilde{\tau}(\tilde{\xi}, x))) \langle \nabla \tilde{\psi}(\tilde{\theta}), \tilde{v} \rangle \right] \right| \tag{65}$$

We bound the right hand side by

$$\left| \langle \nabla_{\tilde{\xi}} \tilde{G}(\tilde{y}_\xi), \tilde{u} \rangle \right| \leqslant \sup_{\hat{f}: \hat{G}(\tau(\xi, x)) = G(\tau(\xi, x))} M^* \max_{\|\tilde{v}\|_2 = 1} \left| \mathbb{E}_{\theta \sim g} \left[ \hat{f}(\tilde{\tau}(\tilde{\theta}, \tilde{\tau}(\tilde{\xi}, x))) \langle \nabla \tilde{\psi}(\tilde{\theta}), \tilde{v} \rangle \right] \right| \tag{66}$$

and the optimal $\hat{f}$ is as follows,

$$\hat{f}(\tilde{\tau}(\tilde{\theta}, \tilde{\tau}(\tilde{\xi}, \tilde{x}))) = \begin{cases} 1, \text{if} \langle \tilde{\psi}(\tilde{\theta}), \tilde{u} \rangle > \varphi_u^{-1}(\tilde{G}(\tilde{\tau}(\tilde{\xi}, \tilde{x}))) \\ 0, \text{else} \end{cases} \tag{67}$$

here

$$M(\xi, \theta) = \begin{bmatrix} I_d, & O_{d \times n} \\ F_{22}'(y_\xi) - F_1'(\theta), & O_{n \times n} \end{bmatrix} \tag{68}$$

and $M^*$ is

$$
\begin{aligned}
M^* &= \max_{\xi,\theta\in P} \left\| \begin{bmatrix} I_d, & O_{d\times n} \\ F_{22}'(y_\xi) - F_1'(\theta), & O_{n\times n} \end{bmatrix} \right\|_2 \\
&= \max_{\xi,\theta\in P} \left\| \begin{bmatrix} I_d \\ \frac{\partial F_{22}(y_\xi)}{\partial \xi} - \frac{\partial F_1(\theta)}{\partial \theta} \end{bmatrix} \right\|_2 \\
&= \max_{\xi,\theta\in P} \sqrt{1 + \left\| \left( \frac{\partial F_{22}(y_\xi)}{\partial \xi} - \frac{\partial F_1(\theta)}{\partial \theta} \right) \right\|_2^2}.
\end{aligned}
\tag{69}
$$

Notice that here $F_{22}(\cdot)$ is the same as $F_2(\cdot)$ the notations in the main text. Then we could apply the techniques used in Theorem 1 and Theorem 4, we have:

$$
R = \frac{1}{2M^*} \int_{\overline{p_B}}^{\underline{p_A}} \frac{1}{\Phi(p)} dp,
\tag{70}
$$

Thus we have proven this Theorem. $\qquad\square$

### A.3   THE PROOF OF THEOREM 3

**Theorem 3.** *Suppose the simulation of the semantic transformation has an small enough error*

$$
\|\tilde{\tau}(\tilde{\xi}, \tilde{x}) - \overline{\tau}(\tilde{\xi}, \tilde{x})\| < \varepsilon,
$$

*Then there exists a constant ratio $A = A(\|F_1'(\tilde{\xi})\|, \|F_2'(\tilde{y}_\xi)\|, \|F_2'(\tilde{z}_\xi)\|) > 0$ does not depend on the target classifier, we have the certified radius for the real semantic transformation satisfies that*

$$
R_r > R(1 - A\varepsilon)
$$

*where $R$ is the certified radius for surrogate the neural network in Theorem 2 and*

$$
R = \frac{1}{2M^*} \int_{\overline{p_B}}^{\underline{p_A}} \frac{1}{\Phi(p)} dp.
$$

*Proof.* We set

$$
\tilde{u} = \begin{pmatrix} u \\ O_{n\times 1} \end{pmatrix},
\tag{71}
$$

here $u \in \mathbb{R}^d$ and $\|u\| = 1$. Then we have

$$
\begin{aligned}
\left\langle \nabla_{\tilde{\xi}} \tilde{G}(\tilde{\tau}(\tilde{\xi}, \tilde{x})) - \nabla_{\tilde{\xi}} \tilde{G}(\overline{\tau}(\tilde{\xi}, \tilde{x})), \tilde{u} \right\rangle &= \int \left( \frac{\partial \tilde{f}(\tilde{\tau}(\tilde{\theta}, \tilde{\tau}(\tilde{\xi}, \tilde{x})))}{\partial \tilde{\xi}} - \frac{\partial \tilde{f}(\tilde{\tau}(\tilde{\theta}, \overline{\tau}(\tilde{\xi}, \tilde{x})))}{\partial \tilde{\xi}} \right) \tilde{u}\tilde{g}(\tilde{\theta}) d\tilde{\theta} \\
&= \int \left( \tilde{\tau}(\tilde{\xi}, \tilde{x}) - \overline{\tau}(\tilde{\xi}, \tilde{x}) \right) \frac{\partial^2 \tilde{f}(\tilde{\tau}(\tilde{\theta}, \hat{\tau}(\tilde{\xi}, \tilde{x})))}{\partial \tilde{\xi} \partial \hat{\tau}} \tilde{u}\tilde{g}(\tilde{\theta}) d\tilde{\theta}
\end{aligned}
\tag{72}
$$

Set $L = \frac{\partial^2 \tilde{f}(\tilde{\tau}(\tilde{\theta}, \overline{\tau}(\tilde{\xi}, \tilde{x})))}{\partial \tilde{\xi} \partial \hat{\tau}}$, we have

$$
L = \frac{\partial}{\partial \tilde{\xi}} \left( \frac{\partial \tilde{f}(\tilde{\tau}(\tilde{\theta}, \hat{\tau}(\tilde{\xi}, \tilde{x})))}{\partial \hat{\tau}} \right) = \frac{\partial}{\partial \tilde{\xi}} \left( \frac{\partial \tilde{f}(\tilde{\tau}(\tilde{\theta}, \hat{\tau}(\tilde{\xi}, \tilde{x})))}{\partial \tilde{\tau}(\tilde{\theta}, \hat{\tau}(\tilde{\xi}, \tilde{x}))} \frac{\partial \tilde{\tau}(\tilde{\theta}, \hat{\tau}(\tilde{\xi}, \tilde{x}))}{\partial \hat{\tau}} \right).
\tag{73}
$$

Set $y_{\tilde{\xi}} = \tilde{\tau}(\tilde{\theta}, \hat{\tau}(\tilde{\xi}, \tilde{x}))$, we have:

$$\frac{\partial \tilde{\tau}(\tilde{\theta}, \hat{\tau}(\tilde{\xi}, \tilde{x}))}{\partial \hat{\tau}} = \frac{\partial \tilde{H}(\tilde{F}_1(\tilde{\theta}) + \tilde{F}_2(\hat{\tau}(\tilde{\xi}, \tilde{x})))}{\partial \hat{\tau}}$$

$$= \tilde{H}'\left(\tilde{F}_1(\tilde{\theta}) + \tilde{F}_2(\hat{\tau}(\tilde{\xi}, \tilde{x}))\right) \frac{\partial \tilde{F}_2(\hat{\tau}(\tilde{\xi}, \tilde{x}))}{\partial \hat{\tau}}$$

$$= \tilde{H}'\left(\tilde{F}_1(\tilde{\theta}) + \tilde{F}_2(\hat{\tau}(\tilde{\xi}, \tilde{x}))\right) \begin{bmatrix} I_d \\ F_1'(\theta) & I_n \end{bmatrix} \begin{bmatrix} I_d \\ -F_1'(\theta) & I_n \end{bmatrix} \frac{\partial \tilde{F}_2(\hat{\tau}(\tilde{\xi}, \tilde{x}))}{\partial \hat{\tau}} \quad (74)$$

$$= \frac{\partial \tilde{\tau}(\tilde{\theta}, \hat{\tau}(\tilde{\xi}, \tilde{x}))}{\partial \tilde{\theta}} \begin{bmatrix} I_d \\ -F_1'(\theta) & I_n \end{bmatrix} \tilde{F}_2{}'(\hat{\tau}(\tilde{\xi}, \tilde{x}))$$

$$= \frac{\partial \tilde{\tau}(\tilde{\theta}, \hat{\tau}(\tilde{\xi}, \tilde{x}))}{\partial \tilde{\theta}} A_1$$

$$\square$$

here $A_1 = \begin{bmatrix} I_d \\ -F_1'(\theta) & I_n \end{bmatrix} \tilde{F}_2{}'(\hat{\tau}(\tilde{\xi}, \tilde{x}))$. By the proof above, we have $\frac{\partial}{\partial \tilde{\xi}} = \frac{\partial}{\partial \tilde{\theta}} A_2$, thus we have:

$$L = \frac{\partial}{\partial \tilde{\xi}} \left( \frac{\partial \tilde{f}(\tilde{\tau}(\tilde{\theta}, \hat{\tau}(\tilde{\xi}, \tilde{x})))}{\partial \tilde{\tau}(\tilde{\theta}, \hat{\tau}(\tilde{\xi}, \tilde{x}))} \frac{\partial \tilde{\tau}(\tilde{\theta}, \hat{\tau}(\tilde{\xi}, \tilde{x}))}{\partial \hat{\tau}} \right)$$

$$= \frac{\partial}{\partial \tilde{\theta}} \left( \frac{\partial \tilde{f}(\tilde{\tau}(\tilde{\theta}, \hat{\tau}(\tilde{\xi}, \tilde{x})))}{\partial \tilde{\tau}(\tilde{\theta}, \hat{\tau}(\tilde{\xi}, \tilde{x}))} \frac{\partial \tilde{\tau}(\tilde{\theta}, \hat{\tau}(\tilde{\xi}, \tilde{x}))}{\partial \tilde{\theta}} A_1 \right) A_2 \quad (75)$$

$$= \frac{\partial}{\partial \tilde{\theta}} \left( \frac{\partial \tilde{f}(\tilde{\tau}(\tilde{\theta}, \hat{\tau}(\tilde{\xi}, \tilde{x})))}{\partial \tilde{\theta}} A_1 \right) A_2.$$

$$\left\langle \nabla_{\tilde{\xi}} \tilde{G}(\tilde{\tau}(\tilde{\xi}, \tilde{x})) - \nabla_{\tilde{\xi}} \tilde{G}(\bar{\tau}(\tilde{\xi}, \tilde{x})), \tilde{u} \right\rangle$$

$$= \int \left( \tilde{\tau}(\tilde{\xi}, \tilde{x}) - \bar{\tau}(\tilde{\xi}, \tilde{x}) \right) \frac{\partial^2 \tilde{f}(\tilde{\tau}(\tilde{\theta}, \hat{\tau}(\tilde{\xi}, \tilde{x})))}{\partial \tilde{\xi} \partial \hat{\tau}} \tilde{u} \tilde{g}(\tilde{\theta}) \mathrm{d}\tilde{\theta}$$

$$= \int \left( \tilde{\tau}(\tilde{\xi}, \tilde{x}) - \bar{\tau}(\tilde{\xi}, \tilde{x}) \right) \frac{\partial}{\partial \tilde{\theta}} \left( \frac{\partial \tilde{f}(\tilde{\tau}(\tilde{\theta}, \hat{\tau}(\tilde{\xi}, \tilde{x})))}{\partial \tilde{\theta}} A_1 \right) A_2 \tilde{u} \tilde{g}(\tilde{\theta}) \mathrm{d}\tilde{\theta} \quad (76)$$

$$\leqslant \tilde{A}\epsilon \cdot \left| \int \tilde{f}(\tilde{\tau}(\tilde{\theta}, \hat{\tau}(\tilde{\xi}, \tilde{x}))) \tilde{g}(\tilde{\theta}) \langle \nabla \tilde{\psi}(\tilde{\theta}), \tilde{u} \rangle \mathrm{d}\tilde{\theta} \right|,$$

where $\tilde{A}$ is a constant depending on $\|F_1'(\tilde{\xi})\|, \|F_2'(\tilde{y}_\xi)\|, \|F_2'(\tilde{z}_\xi)\|$. Then there exists $A$ and we have

$$R_r \geqslant R(1 - \epsilon A), \quad (77)$$

where

$$R = \frac{1}{2M^*} \int_{\overline{p_B}}^{\overline{p_A}} \frac{1}{\Phi(p)} dp$$

# B  IMPLEMENTATION DETAILS AND EXPERIMENTAL SETTINGS

## B.1  PRACTICAL ALGORITHMS FOR CALCULATING $M^*$

For resolvable transformations in Theorem 1, the $M^*$ is defined as

$$M^* = \max_{\xi, \theta \in P} \|M(\xi, \theta)\|. \quad (78)$$

Since if we have verified that the semantic transformation is resolvable, most of time we have a closed form of $M^*$ like contrast/brightness transformation and we are able to calculate it analytically as shown in Li et al. (2021).

For non-resolvable transformations in Corollary 1, $M^*$ is defined as

$$M^* = \max_{\xi \in P} \sqrt{\frac{1}{\sigma_1^2} + \frac{1}{\sigma_2^2} \left\| \frac{\partial F_2(y_\xi)}{\partial \xi} - A_1 \right\|_2^2}. \tag{79}$$

This ratio is similar with the Lipschitz bound for a semantic transformation in Li et al. (2021).For low dimensional semantic transformations, we are able to interpolate the domain to find a maximum $M^*$ and corresponding $\xi$. But this remains a challenge for high dimensional semantic transformations. Specifically, for a given $\xi$, we need to compute the norm of $\frac{\partial F_2(y_\xi)}{\partial \xi} - A_1$. The Jacobian matrix is $n \times n$. Caculating it requires $n$ times of backpropagation. Thus it is inefficient to store the matrix or directly compute its norm. To solve the problem, we noitice that

$$\left\| \frac{\partial F_2(y_\xi)}{\partial \xi} - A_1 \right\|_2^2 = \max_{\|u\|_2=1} \left\| \left( \frac{\partial F_2(y_\xi)}{\partial \xi} - A_1 \right) u \right\|_2^2. \tag{80}$$

And then we have

$$\left( \frac{\partial F_2(y_\xi)}{\partial \xi} - A_1 \right)^T u = \frac{\partial}{\partial \xi} \langle F_2(y_\xi) - A_1, u \rangle \tag{81}$$

Since tranposing a matrix does not change its norm, we could calculate its norm by optimizing $u$ that,

$$\max_{\|u\|_2=1} \left\| \frac{\partial}{\partial \xi} \langle F_2(y_\xi) - A_1, u \rangle \right\|_2^2. \tag{82}$$

Using this formulation, we only need to multiply the output with an unit vector and perform one backpropagation. This is a simple convex optimization problem. Then we could use any iterative algorithm to find its solution which is very fast to compute. This trick is crucial and it makes the matrix norm computation to be scalable in practice.

### B.2 OTHER DETAILS FOR EXPERIMENTS.

Our GSmooth requires to train two neural networks. First we randomly generate corrupted images to train a image-to-image neural network. The training process of classifiers and certification for semantic transformations are done on 2080Ti GPUs. We use a U-Net (Ronneberger et al., 2015) for the surrogate model and replace all BatchNorm layers with GroupNorm (Wu & He, 2018) since we might use the model in low bacthsize settings. The U-Net could be replace by other networks used in image segmentation or superresolution like Res-UNet (Diakogiannis et al., 2020) or EDSR (Lim et al., 2017). We use L1-loss to train the surrogate model which achieves better accuracy than others which is also reported (Lim et al., 2017).

After train a surrogate model to simulate the semantic transformation. Then we train the base classifier for certification with a moderate data augmentation like (Li et al., 2021; Cohen et al., 2019) to ensure that training and testing of the classifier is performed on the same distribution. There are two types of data augmentation, one is the semantic transformation and the other is the augmented noise introduced only in our work. Data augmentation based semantic transformation could be done using both the surrogate model or the raw semantic transformation. We can only use the surrogate model to add the augmented noise because this noise is a type of semantic noise in the layers of surrogate model. In our experiments the standard deviation of the augmented noise is chosen from $0.1 \sim 0.4$ depending on the performance. The basic network architectures for these datasets are kept the same with (Li et al., 2021). On CIFAR-100 daatsets, we use a PreResNet (He et al., 2016b) re-implement the method by Li et al. (2021). We also adopt the progressive sampling trick mentioned in TSS (Li et al., 2021) which is useful to reduce computational cost and certify larger radius. The details could also be found in Li et al. (2021).

## C SUPPLEMENTARY EXPERIMENTS

In this section, we report the results of our GSmooth under adaptive attacks to verify the tightness of our certified bound. The experiments are conducted on CIFAR-10 dataset. We use expectation of transformation to calculate the gradient of the model. Then we apply projected gradient descent

|  | Cert Acc(%) | EoT attacks(%) |
|---|---|---|
| Gaussian blur | 67.4 | 68.1 |
| Tanslation | 82.2 | 87.5 |
| Rotation | 64.6 | 68.4 |
| Rotational blur | 39.7 | 45.0 |
| Defocus blur | 25.0 | 25.0 |
| Pixelate | 45.3 | 49.2 |

Table 3: Accuracy of our GSmooth under adaptive attacks (PGD using expectation over transformations) on CIFAR-10 dataset.

| Type/Acc(%) | Augmix | TSS | Ours |
|---|---|---|---|
| Zoom blur | 70.8 | 75.2 | **77.1** |
| Defocus blur | 72.2 | 75.6 | **76.8** |
| Pixelate | 50.9 | 76.0 | **76.7** |
| Brightness | 82.4 | 71.8 | **72.1** |
| Motion blur | 68.6 | 70.2 | **70.5** |
| Gaussian blur | 67.4 | **75.8** | 75.2 |

Table 4: Empirical accuracy on subsets of CIFAR-10-C.

to find adversarial examples until it converged. Then we report the accuracy of our model on the corrupted dataset. The result is listed in the following table. We found that the empirical attack is no less than the certified accuracy. This shows that the bound for our model is effective. Additional, some empirical results are much higher than the certified accuracy, which might indicate the bound still has space for improvement.

We have conducted some experiments under the common benchmark CIFAR-10-C to show the empirical robustness under these corruptions. We conducted the experiments on some subsets of CIFAR-10-C. The corruption types of these subsets are related to the experiments in our paper. The settings of these experiments and the results of baselines are from TSS(Li et al 2021).

