# OpenReview forum: "GSmooth: Certified Robustness against Semantic Transformations via Generalized Randomized Smoothing"
_ICLR.cc/2022/Conference — ICLR 2022 Submitted_

### Official Review · Reviewer_bwyb · 2021-11-01

**Correctness:** 3
**Technical Novelty And Significance:** 4
**Empirical Novelty And Significance:** 4
**Recommendation:** 8
**Confidence:** 5

**Main Review:**

In my opinion, this work shows ground-breaking results in certified robustness, by providing an elegant and general framework for robustness certification with arbitrary and possibly complex semantic transformations with performance guarantees. I believe it achieves an important milestone in adversarial robustness.

The paper is also well-written and easy to follow. The comparisons to existing methods are thorough and convincing. The ablation studies (especially the noise distribution part that uniform distribution is quite competitive) also shed some new insights into randomized smoothing.

I have two suggestions that I hope the authors can incorporate into the revised version:

1. In Table 1, since the considered transformation has very few parameters, I would like to see the comparison of empirical robustness by running the actual attack on the certified value to understand the gap between certified and empirical robustness.

2. Since the proposed method can handle composite semantic transformations, can the authors some case studies to demonstrate such an advantage?

Minor comment"
1. In Sec. 3.1, \nabla should be defined, and I believe \psi function needs to be differentiable


**Summary Of The Paper:**

This paper proposed GSmooth, a generalized randomized smoothing method for semantic transformations.

The main technical contributions are:
(1) Introduce the use of an image-to-image translation network to provide a unified framework for the analysis of non-resolvable semantic transformations.

(2) Theoretical proof on the certified radius and the approximation error from the image-to-image translation network.

(3) The empirical performance is superior to existing methods on most transformations. More importantly, the method can certify many new transformations that are hard to analyze based on existing methods.


**Summary Of The Review:**

1. Strong theoretical and empirical results
2. The contributions are significant for expanding randomized smoothing to more complex semantic transformations

---

> ### Author Response · Authors · 2021-11-11
> **Thank you for your supportive review**
>
> Thank you for appreciating our idea and approach, as well as the valuable suggestions. Below we address the detailed comments. We have conducted some experiments on the empirical accuracy of our GSmooth under adaptive attacks(Projected Gradient Descent + Expectation over Transformation for gradient calculation). We first use the CIFAR-10 dataset and the results are shown in the following table. The attack range of these parameters are kept the same with our main experiments.
>
> | Type          | Cert Acc | Adaptive Attack Acc |
> | ---------------| | -------- |
> | Gaussian blur   |67.4%| 68.1% |
> | Translation     |82.2%| 87.5% |
> | Rotation        |64.6%| 68.4% |
> | Rotational blur | 39.7% | 45.0% |
> | Defocus blur    | 25.0% | 25.0% |
> | Pixelate        | 45.3% | 49.2% |
>
> **Q1**: In Table 1, since the considered transformation has very few parameters, I would like to see the comparison of empirical robustness by running the actual attack on the certified value to understand the gap between certified and empirical robustness.
>
> **A1**:  We have conducted experiments that show the empirical robustness of GSmooth and results will be updated in revised version.
> The key results are shown in the table above.
>
> **Q2**: Since the proposed method can handle composite semantic transformations, can the authors some case studies to demonstrate such an advantage?
>
> **A2**:  Thanks for your good advice. We will provide more case studies in the revised version.
>
> **Q3**: In Sec. 3.1, $\nabla$ should be defined, and I believe $\psi$ function needs to be differentiable.
>
> **A3**: Yes, we have corrected them in the revised version.

---

> ### Comment · Reviewer_bwyb · 2021-11-22
> **Thank you authors for the prompt updates**
>
> I thank the reviewer for providing all the suggested experiments and preliminary results. These results make the proposed certification method even more convincing. I believe including those results will further strengthen the contributions of this paper.

---

### Official Review · Reviewer_HKx4 · 2021-11-01

**Correctness:** 2
**Technical Novelty And Significance:** 3
**Empirical Novelty And Significance:** 2
**Recommendation:** 3
**Confidence:** 3

**Main Review:**

In this work, the authors first construct a smoothed classifier $G(x)$, based on smoothing the parameter of the parametrised transformation $\tau$. Based on the compositional properties of $\tau$, they define conditions to obtain a robustness radius (Theorem 1). Then they proceed to construct a surrogate image to image translation model approximating the transformation $\tau$. Then the parameters of $\tau$ and inputs $x$ get augmented to $\tilde{\tau}$ and $\tilde{x}$ respectively to divide the transformation into a resolvable and a non resolvable part. The surrogate transformation then leads to a more concrete form of $M^*$ (Theorem 2), which is further simplified by assuming that $F_1$ is an affine form (Corollary 1). Finally, in Theorem 3, the authors provide a correction for the robustness radius $R_r$ for the case where surrogate transformation $\tilde{\tau}$ is approximating $\bar{\tau}$ dependent on the maximum $\ell_2$ approximation error.

The main ideas of the paper are interesting and to the best of my knowledge novel, particularly the surrogate network. However, this paper needs to be written much more clearly to convey the central ideas to the reader. Particularly Section 3 and Section 4 are hard to understand and required considerable effort, after which some aspects remain still unclear. On page 5: What are the dimensions for $x’$ and $\theta’$ or $\tilde{x}, \tilde{\theta}$ and how is $x’$ chosen? How do you decompose the transformation to get a resolvable and a non resolvable part? Do i guess correctly from Eq. 9, that the resolvable part is the sum between $\theta$ and $x’$, and the non resolvable part is the sum between $F_1(\theta) + \theta’$ and $F_2(x)$?

The related work Li et al 2021 and Fischer et al 2020 do evaluate on ImageNet. Can the same be done for GSmooth? Table 1 indicates that VeriVis is not able to certify against translations. However, VeriVis can certify translations. Was the certification accuracy 0? What about the other works? Can the other methods potentially be adapted to handle Blur or the Pixelate transformation? Further, the runtimes need to be stated in order to judge the trade off between performance and accuracy. Where do the authors show how they calculate their $\epsilon$ for the surrogate transformation networks as needed for Theorem 3?

The authors claim multiple times (abstract and later in the paper) that existing work can not handle complex semantic transformations. Can the authors substantiate this impossibility? I could not find a proof or some further justification for this. Further, in the related work, section 2.1: Apart from Madry et al. 2017, these works do not attempt to certify some simple geometric transformations but actually do certify them. Mardy et al 2017 on the other hand focuses on attacks and defenses. The authors clarify the difficulties the approach of Li et al 2021 presents for generic transformations. Can the authors also clarify the difficulties the approach of Fischer et al. 2020 presents? In Section 3.3. the authors imply that existing methods based on convex relaxations and randomized smoothing require the development of a specific algorithm for each individual transformation. To what extend is this true for Balunovic et al 2019 and Fischer et al 2020?

Minor comments:
- Eq. 1: It is slightly confusing to write $\theta \sim g(\theta)$.
- Just before Eq. 8: the dimensions for $\tilde{\tau}$ seem off.

**Summary Of The Paper:**

This paper generalizes randomized smoothing to certify robustness against complex semantic transformations. To do that, they construct a surrogate neural network mapping images to images, to approximate complex semantic transformations and certify robustness with respect to this network.

**Summary Of The Review:**

While the ideas in this paper are to the best of my knowledge interesting and novel, many questions remain unanswered and key points in the paper remain unclear.

---

> ### Author Response · Authors · 2021-11-12
> **Thank you for your valuable review (Part 1)**
>
> Thank you for appreciating our idea and approach, as well as the valuable suggestions. Below we address the detailed comments. We have conducted some experiments on the empirical accuracy of our GSmooth under adaptive attacks(Projected Gradient Descent + Expectation over Transformation for gradient calculation). We first use the CIFAR-10 dataset and the results are shown in the following table. The attack range of these parameters are kept the same with our main experiment.
>
> | Type          | Cert Acc | Adaptive Attack Acc |
> | ---------------| | -------- |
> | Gaussian blur   |67.4%| 68.1% |
> | Translation     |82.2%| 87.5% |
> | Rotation        |64.6%| 68.4% |
> | Rotational blur | 39.7% | 45.0% |
> | Defocus blur    | 25.0% | 25.0% |
> | Pixelate        | 45.3% | 49.2% |
>
> **Q1**:  On page 5: What are the dimensions for $x'$ and $\theta'$ or and how is $\tilde{x}$ and $\tilde{\theta}$ chosen?
>
> **A1**: $x$ is the original data and its dimension is $n$, $\theta$ is the transformation parameter and its dimension is $m$. $x'$ is the ``augmented'' data which can be set as a zero array with dimension $d-n$. The dimension of augmented transformation parameter $\theta'$ is $d-m$. Actually, we only require to augment $\theta$, the augmentation of data $x$ is to keep the dimensions to be consistent in our theoretical results. We have added more clear explanations to them in the revised version.
>
> **Q2**:  How do you decompose the transformation to get a resolvable and a non-resolvable part?
>
> **A2**:  The decomposition is not an explicit process but is an intuitive understanding of the key steps of our theoretical proof in Eq. (50) in the appendix. The decomposition happens when we introduce the augmented noise in Eq. (6) of the main text. An intuitive explanation is that a semantic transformation modifies an image along a low dimensional subspace in the manifold. Our insight is to use a surrogate neural network designed in our paper to **approximate the manifold** as stated in our paper. Then components on the manifold can be certified using a transformation-based randomized smoothing. The residual components which are the **non-resolvable parts have to be tackled using the augmented noise** that fills the whole latent space. And it is not right to understand the resolvable part as the $x'+\theta$.
>
> **Q3**:  Is the resolvable part $x'+\theta$? The non-resolvable part is $F_1(\theta)+\theta'+F_2(x)$?
>
> **A3**:  Actually not. This is a conceptual explanation for the key steps of GSmooth. If we have to divide it explicitly, an informal explanation is that the noise $F_1(\theta)$ bounds the resolvable part, and the augmented noise $\theta'$ bounds the non-resolvable part. We have added more explanations in the revised version.
>
> **Q4**: The related work Li et al 2021 and Fischer et al 2020 do evaluate on ImageNet. Can the same be done for GSmooth?
>
> **A4**: Our work is theoretically applicable for ImageNet dataset. But there is a problem in practice. The sizes of ImageNet is usually 224*224. Training an accurate image-to-image translation model from scratch on ImageNet is much more difficult than on CIFAR-10. We have attempted to do so and progress will be updated in future works.
>
> **Q5**: Table 1 indicates that VeriVis is not able to certify against translations. However, VeriVis can certify translations. Was the certification accuracy 0?
>
> **A5**: In Table 1, the certified accuracy of VeriVis on translation is *98.8\%, 65.0\%, 24.2\%* and they are not 0. The results are consistent with *Li et al 2021*. on MNIST and CIFAR-10.
>
> **Q6**: What about the other works? Can the other methods potentially be adapted to handle Blur or the Pixelate transformation?
>
> **A6**: It is a very difficult task to derive a certified bound for these complex semantic transformations, which is also mentioned in Li et al 2020.  We will show a comparison between all these related works in the next part.
>
> **Q7**: Further, the runtimes need to be stated in order to judge the trade-off between performance and accuracy.
>
> **A7**: The additional cost of our method is to train a surrogate model and this could be done offline. So it does not bring much computational load when certifying because the forward pass of the surrogate model is fast.
>
> **Q8**:  Where do the authors show how they calculate their $\epsilon$ for the surrogate transformation networks as needed for Theorem 3?
>
> **A8**: The error constant cannot be calculated precisely but the order of magnitudes is about 1e-3 in our experiments. The error for fitting a semantic transformation is usually less than 1e-4, and the term about the norm of $M^\ast$ is about 1~10 in experiments. So the total error term is less than 1e-3 in practice and this does not hurt the certified accuracy in experiments. We will provide more detailed results in the revised revision.

---

> > ### Author Response · Authors · 2021-11-12
> > **Thank you for your valuable review (Part 2)**
> >
> > **Q9**: The authors claim multiple times (abstract and later in the paper) that existing work can not handle complex semantic transformations. Can the authors substantiate this impossibility?  The authors clarify the difficulties the approach of Li et al 2021 presents for generic transformations. Can the authors also clarify the difficulties the approach of Fischer et al. 2020 presents? In Section 3.3. the authors imply that existing methods based on convex relaxations and randomized smoothing require the development of a specific algorithm for each individual transformation. To what extend is this true for Balunovic et al 2019 and Fischer et al 2020?
> >
> > **A9**:  We divide all these existing works into two types and discuss them respectively. We compare these methods in the following table in summary and then discuss them in detail.
> >
> > | Method                              | Assumption                                                  | Shortcomings                                                |
> > | ----------------------------------- | ----------------------------------------------------------- | ----------------------------------------------------------- |
> > | TSS (Li et al 2021)                 | differentiable resolvable                                   | Calculate individual bound for each transformation , not scalable to more complex transformations         |
> > | distSPT(Fischer et al. 2020)        | **additive**  transformation                                    | Analysis error bound for only **rotation/translation** due to interpolation      |
> > | DeepG (Balunovic et al 2019)        | combination of **bijective** transformations and interpolations | Not applicable to some blur-based transformations           |
> > | Semanify-NN(Mohapatra et al., 2020) | Transform the semantic certification into l-p certification | Only applicable to several specific preset transformations  |
> > | VeriVis (Pei et al., 2017)          | enumeration based                                           | Only available for transformations with discrete parameters |
> >
> >
> > 1. The first class is Li et al 2021 (TSS in baselines), Fischer et al. 2020 (distSPT) based on randomized smoothing. TSS (Li et al 2021) is able to certify some simple geometric transformations like rotation, translation and it requires **a specific Lipschitz bound** for each non-resolvable transformation as mentioned in the paper. distSPT(Fischer et al. 2020) has an assumption that the semantic transformation **must be additive** which is stated as a subsection in our paper. distSPT (Fischer et al 2020) then uses a lot of text to calculate **a specific error bound for rotation** transformation due to the additive approximation. The inverse computation it requires is not applicable for many blur-based transformations which are also mentioned by TSS (Li et al 2021).
> >
> > 2. The second class is DeepG (Balunovic et al 2019), Semanify-NN (Mohapatra et al 2020) and VeriVis (Pei et al., 2017) that are deterministic approaches. DeepG (Balunovic et al, 2019) assumes the geometric transformations are written in **a combination of a bijective transformation, an interpolation, and a contrast**. Then it calculates a convex relaxation of all possible inputs. This is a strong assumption that does not hold for complex semantic transformations like blur or pixelate that is not bijective. Semanify-NN (Mohapatra et al., 2020) and Interval (Singh et al., 2019) also adopt similar approaches so they are also not applicable for complex semantic transformations. VeriVis (Pei et al., 2017) models the problem as **a search problem and uses enumeration-based certification**. Only discrete cases like translation could be certified which is also indicated by Li et al 2021.
> >
> > In a summary, these prior works are able to certify some simple semantic transformations. But they are still limited because they have **strong assumptions** which are not scalable to more complex semantic transformations.  Our main contribution is to **relieve some of these assumptions** which enables us to certify a wider range of semantic transformations.
> >
> > References:
> >
> > 1.  https://arxiv.org/abs/2002.12398 TSS: Transformation-Specific Smoothing for Robustness Certification
> > 2.  https://arxiv.org/abs/2002.12463 Certified Defense to Image Transformations via Randomized Smoothing
> > 3.  https://arxiv.org/abs/1912.09533 Towards Verifying Robustness of Neural Networks Against Semantic Perturbations
> > 4.  https://arxiv.org/abs/1712.01785 Towards Practical Verification of Machine Learning: The Case of Computer Vision Systems
> > 5.  https://papers.nips.cc/paper/2019/file/f7fa6aca028e7ff4ef62d75ed025fe76-Paper.pdf  Certifying Geometric Robustness of Neural Networks

---

> > > ### Comment · Reviewer_HKx4 · 2021-11-18
> > > **Follow up questions**
> > >
> > > Thank you for your detailed and helpful response. I just have a few follow up questions:
> > >
> > > Q5: Sorry, here i meant rotations for VeriVis. Further, to the best of my knowledge, DeepG, Interval and IndivSPT/DistSPT can handle translations. What is “-” indicating there?
> > >
> > > Q8: How where the numbers obtained? Is this a theoretical upper bound? Can the surrogate model be adversarially attacked?

---

> ### Author Response · Authors · 2021-11-26
> **A gentle reminder to reviewers**
>
> Dear reviewer  HKx4,
>
> Thank you again for your valuable comments and suggestions, which are really helpful for us. We have posted responses to the detailed concerns.
>
> We totally understand that this is a quite busy period, since the reviewers may be responding to the rebuttal of other assigned papers.
>
> We deeply appreciate it if you can take some time to return further feedback on whether our responses solve your concerns. If there are any other comments, we will try our best to address them.
>
> Best,
>
> The authors

---

### Official Review · Reviewer_r3hs · 2021-11-02

**Correctness:** 3
**Technical Novelty And Significance:** 3
**Empirical Novelty And Significance:** 3
**Recommendation:** 8
**Confidence:** 4

**Main Review:**

Strengths:
1. The topic of certifying generalized semantic transformations is extremely relevant. The paper is well motivated and the authors clearly point out differences with existing work.
2. GSmooth is able to certify several non-trivial semantic transformations which include difficult combinations such as rotational and defocus blur.
3. While the certified accuracy values are not that significant for the more complex transforms, the algorithm is good first attempt which succeeds as compared to other methods.

Weaknesses:
1. I am not convinced that an $\ell_2$ bound in the parameter space for compound transformations makes sense. A real world attacker instead would have $\ell_\infty$ or $\ell_2$ type bounds for parameters of each transform. For example in the case of rotational blur, an attacker could independently increase rotations and blur to the max value. Given the very different ranges of the two tranformations, an $\ell_2$ bound may emphasize one transform over the other. The current algorithm does not seem to take this into consideration.
2. The idea of augmenting the latent space to decouple resolvable and non-resolvable components is interesting. However, from the text, it is not clear why such a decoupling would occur without additional training objectives for the surrogate. Could the authors clarify this?
3. It's not clear to me why H would not reflect in the M* term. The proof seems to ignore the effect of H by considering it to be a concatenation of an identity and a zero matrix. However, the decoder itself may not be Lipschitz due to the non-trivial nonlinearities involved. A more clear explanation here is warranted.
4. The paper lacks a empirical confirmation of the presented bound. Specifically, the GSmooth classifier should be tested against an adaptive attack that has access to the surrogate and the parameter space to find if the bounds do actually hold. The certificate should also be tested against standard Expectation over Transformation attacks. This would also help quantify the tightness of the proposed bound.

Minor comments:
1. The explanation regarding the dimensionality augementation is confusing and can be presented better-- "Specifically, we introduce the augmented data $\tilde{x}$ ... additional dimensions to 0.''. Also, Eq. 9 should include the expression for augmented $\tilde{H}$ for better clarity.
2. Table 2 does not mention the radius for the certified accuracies.
3. Section 3 needs to be more clear. The notation for the augmented parameter space needs some clarity on the dimensions used.
4. While the authors present a nice simplification for F_1, how about explicitly forcing lipschitzness for F_1, F_2, and H?

**Summary Of The Paper:**

The authors propose a randomized smoothing based certification algorithm  for general semantic transformations. The key idea of the work is to use a neural surrogate for the semantic transformations, and add noise in the latent space of the surrogate for randomized smoothing. The neural surrogate appears to convert the non-linear, multiplicative transformation into an additive operation in the latent space, thus allowing for randomized smoothing methods to be applied. Their approach proposes to decompose the resolvable and unresolvable parts of the semantic transformation by lifting the data+transformation parameters into a larger augmented latent space defined by an image-to-image network. The resolvable parts of the transform can then be certified similar to previous works (Yang'2021, Salman'2019) using the Lipschitzness of the smoothed transforms. The non-resolvable part of the transform however are assumed to be Lipschitz in the latent space. The authors provide theoretical and empirical evidence for GSmooth and show improvement on contemporary methods.

**Summary Of The Review:**

Overall, the paper presents a novel and interesting addition to the randomized smoothing literature. This appears to be the first attempt at certifying complex semantic transformations, and therefore has merit. The paper still needs some additional work however, specifically in terms of quantifying the tightness of their proposed bound against semantic attacks. I also believe the authors should take a more careful look at the actual certificates for the compound attacks. An $ell_2$ bound for the entire parameter vector may not capture the worst case for each component of the transform.

The paper in its current form is marginally below the acceptance threshold. However, if the authors show empirical evidence of their certificates against semantic attacks, I will be glad to improve my score.

---

> ### Author Response · Authors · 2021-11-11
> **Thank you for your valuable review**
>
> Thank you for appreciating our idea and approach, as well as the valuable suggestions. Below we address the detailed comments.
> We have conducted some experiments on the empirical accuracy of our GSmooth under adaptive attacks(Projected Gradient Descent + Expectation over Transformation for gradient calculation). We first use the CIFAR-10 dataset and the results are shown in the following table. The attack range of these parameters are kept the same with our main experiment.
>
> | Type          | Cert Acc | Adaptive Attack Acc |
> | ---------------| | -------- |
> | Gaussian blur   |67.4%| 68.1% |
> | Translation     |82.2%| 87.5% |
> | Rotation        |64.6%| 68.4% |
> | Rotational blur | 39.7% | 45.0% |
> | Defocus blur    | 25.0% | 25.0% |
> | Pixelate        | 45.3% | 49.2% |
>
> **Q1**: I am not convinced that an $l_2$  bound in the parameter space for compound transformations makes sense. A real world attacker instead would have $l_\infty$ or $l_2$ type bounds for parameters of each transform.
>
> **A1**: This is a good question and it has been studied in prior works (Greg Yang et al.). The certified radius for $l_2$ norm is about $\sqrt{n}$ bigger than for $l_\infty$ norm with $n$ being the dimensionality. The certified bounds are also applicable to $l_\infty$ norm and proofs are nearly the same. But notice here $n$ is the **number of freedoms of semantic transformations** rather than the **data dimensions**. For instance, if you have a semantic transformation that contains $4$ parameters, like 2 for translation and 2 for blurring, the certified radius for $l_\infty$ is about 1/2 for $l_2$ norm. As for the scale, transformations that have a larger range, like translations, usually can tolerate higher levels of noise. This normalizes the ranges of semantic transformations.
>
> **Q2**: However, from the text, it is not clear why such a decoupling would occur without additional training objectives for the surrogate.
>
> **A2**: The decoupling of resolvable and non-resolvable parts happens by **introducing the augmented noise in latent space** in Eq.(6) and Eq.(50) which is the key of our theoretical analysis. An intuitive explanation is that a semantic transformation modifies an image along a low dimensional subspace in the manifold. Our insight is to use a surrogate neural network designed in Eq.(8) in our paper to approximate the manifold. Then components on the manifold can be certified using a transformation-based randomized smoothing. The residual components which are the non-resolvable parts have to be tackled using the augmented noise that fills the whole latent space. More formally, the augmented noise **makes the Jacobian matrix of transformation invertible** and details can be found in Eq.(50) of the proof.
>
> **Q3**:  Why H would not reflect in the M* term?
>
> **A3**: I think this is the magic of randomized smoothing from (Jeremy M Cohen et al.). Randomized smoothing itself does not assume **any** properties of the classifier. Here in our method, the whole classifier is $f(H(F_1(x)+F_2(\theta)))$, we could view $f(H(.))$ as a whole. The $M\ast$ depends on $F_1$ and $F_2$ that map the images and transformation parameters to the latent space. But the results of randomized smoothing show that this does not depend on the classifier $f(H(.))$. This happens when we integrate on the noise space and we could eliminate the gradient on $f(H(.))$ similar to original randomized smoothing.
>
> **Q4**:  However, the decoder itself may not be Lipschitz due to the non-trivial nonlinearities involved. A more clear explanation here is warranted.
>
> **A4**: It can also be explained by the results that randomized smoothing does not make any assumptions on the classifier. The nonlinearities of the decoder do not influence our results.
>
> **Q5**: The paper lacks a empirical confirmation of the presented bound.
>
> **A5**: Thanks for your advice. We have conducted experiments on empirical results for adaptive attacks and attacks using EOT. **More results will be updated soon.**
>
> **Q6**:  About the presentation.
>
> **A6**: Thanks for your advice, we will use better notations and add more explanations and it will be updated soon.
>
> **Q7**: Table 2 does not mention the radius for the certified accuracies.
>
> **A7**: The horizontal axis is the certified radius. It should be the $l_2$ norm of $\alpha$ and this experiment is done on zoomed blur. We have corrected them in the revised version.
>
> **Q8**:  How about explicitly forcing lipschitzness for $F_1, F_2$, and $H$?
>
> **A8**: This is OK if we have an explicit constraint on them and this makes our results simpler. But it is a nontrivial problem to design a neural network that has an explicit Lipschitz constant, which remains an open problem for adversarial robustness.
>
> References:
> 1. https://arxiv.org/abs/2002.08118 Randomized Smoothing of All Shapes and Sizes.
> 2. https://arxiv.org/abs/1902.02918 Certified Adversarial Robustness via Randomized Smoothing.

---

> > ### Comment · Reviewer_r3hs · 2021-11-12
> > **Some queries**
> >
> > Thank you for the enlightening response! It does clarify a few of the issues I had. I had a question about the adaptive attacks-does your attack use the original transformation function for calculating EoT, or does it use your trained surrogate?

---

> > > ### Author Response · Authors · 2021-11-13
> > > **Thank you for your response**
> > >
> > > Thank you again for your feedback.
> > >
> > > We use the original transformations to calculate the gradient to attack the model using EoT.
> > >
> > > But notice that our model is a smoothed classifier, so it requires samples from a noise distribution and average their scores. In this step, we use our surrogate model to get corrupted images because we need it to calculate $M^\ast$ in Eq. (11) in our theorem 2.

---

> > ### Comment · Reviewer_r3hs · 2021-11-22
> > **Updating my review**
> >
> > I thank the authors for clarifying my queries. I also would like to commend them for being active and prompt during the discussion period. I believe the authors have satisfactorily addressed my concerns regarding the empirical verification of the proposed bounds as well as testing the certificates against adaptive attacks. While I am still not convinced completely regarding the use of an $\ell_2$ budget for a non-resolvable complex transform, I think this will be a good addition to the conference and of interest to researchers working on adversarial defenses. I am therefore updating my score to 8 (Accept).

---

> > > ### Author Response · Authors · 2021-11-23
> > > **Thanks for the update**
> > >
> > > Dear reviewer r3hs,
> > >
> > > Thank you very much for increasing the score and valuable comments. We'll try our best to further improve the paper and all new experimental results will be updated in the final version.

---

### Official Review · Reviewer_oUek · 2021-11-04

**Correctness:** 4
**Technical Novelty And Significance:** 2
**Empirical Novelty And Significance:** 3
**Recommendation:** 5
**Confidence:** 3

**Main Review:**

Overall, I would recommend a reject for this paper.

The paper discusses an important topic (robustness to semantic transformations), but does not tackle the problem from a perspective that I feel is significant. In particular, it is unclear to me how useful certified robustness to semantic transformations will be, especially for non-resolvable transformations. What does it mean to be robust to rotational blur for attack range ||alpha||_2 < 20 for CIFAR-10, for example? Does this correspond to any guarantees on real datasets, or at least empirical improvements on benchmark datasets?

I appreciate that the authors are thorough with comparing to various prior works in their experiments. However, their method usually performs very similarly or slightly worse than prior works when comparisons are possible; thus, the main experimental improvement is that it is now possible to achieve results on certain non-resolvable transformations that prior works are incapable of handling.

Still, in my view, a key question is unanswered - how can we be sure that the generative model is capturing all the possible non-resolvable transformations that we may care about? Because if we can not answer this question, then it seems that certified robustness is not particularly meaningful, and instead, empirical robustness on benchmark datasets (for example, measuring accuracy on CIFAR-C as well) would be a better measure of success.

Additionally, the paper could be presented better; in particular, the important components should be explained more clearly (e.g. the question I asked above about what ||alpha||_2 < 20 means), whereas other parts of the paper can be moved to the Appendix. For example, Theorems 1 and 2 and Corollary 1 are primarily based on prior work and use up a lot of text, but there is not much description or explanation of how these theorems can be applied to get certified robustness to semantic transformations. The best example of a useful explanation that I found is the short paragraph just under Figure 1 on page 4, which was helpful for me.

Additionally, I believe the authors are missing very relevant prior work. The work of [1] discusses a similar idea, where semantic transformations are modeled via generative models, and Lp-robustness in the latent space of the generative model is used as a proxy for robustness to semantic transformation.

Some more specific comments and questions:
- Can the notation section be simplified any further?
- How large is the error term epsilon in Theorem 3 look for the generative models used in the experiments?

[1] https://arxiv.org/abs/2005.10247 Model-Based Robust Deep Learning: Generalizing to Natural, Out-of-Distribution Data (Alexander Robey, Hamed Hassani, George J. Pappas)




=======================================================================

After the rebuttal period, I have read the author's responses, and changed my score from a 3 to a 5. I appreciate the authors taking the time to attempt to respond to the concerns of all reviewers, and for updating and improving their work during the rebuttal process.

I am glad to see that they do provide empirical evidence of improvement to common-corruption robustness, compared to AugMix (one of the state-of-the-art approaches for standard common-corruption robustness) and TSS, although I can not tell how the authors derived their baselines (I can not find references to the AugMix accuracy numbers that the authors provided in their rebuttal in the TSS or AugMix paper).

Still, in my opinion the paper's novelty is limited. As the authors and I agreed upon in our discussion, the main novelty is not improvement for resolvable transformations (prior works that the authors cite perform about the same or better), but rather, is the ability to handle non-resolvable transformations. I agree that robustness to non-resolvable transformations is important; however, certified robustness to non-resolvable transformations is not meaningful to me, because they are only being certified with respect to a neural network that is trained to approximate those non-resolvable transformations. Without MTurk studies to confirm how good the neural network's non-resolvable transforms are, I do not find certified robustness here meaningful, because it does not necessarily correspond to anything concrete that we can understand. On the other hand, empirical improvements on non-resolvable transformations would be meaningful.

Thus, the main reason I increased my score to weak reject is due to slight empirical improvements over baselines on CIFAR-10C.

**Summary Of The Paper:**

This paper proposes a more generalized form of certified robustness and attempts to provide new results on applying randomized smoothing to semantic transformations such as different types of blurs or distortions. The main idea is to use an image-to-image neural network to approximate semantic transformations, and then certify robustness based on bounds on that neural network. The authors provide empirical results on standard datasets like MNIST and CIFAR showing that their method can achieve improved results on some transformations compared to prior work.

**Summary Of The Review:**

The paper leaves some key questions unanswered, and can be presented much more clearly. It is also missing very relevant prior work. Thus, I recommend a reject.

---

> ### Author Response · Authors · 2021-11-11
> **Thank you for your valuable review (Part 1)**
>
> Thank you for appreciating our idea and approach, as well as the valuable comments. Below we address the detailed comments.
> We have tested our model under adaptive attacks(PGD+EOT) using the CIFAR-10 dataset to verify the tightness of the bound. The empirical accuracy is listed in the table below.
>
> | Type          | Cert Acc | Adaptive Attack Acc |
> | ---------------| | -------- |
> | Gaussian blur   |67.4%| 68.1% |
> | Translation     |82.2%| 87.5% |
> | Rotation        |64.6%| 68.4% |
> | Rotational blur | 39.7% | 45.0% |
> | Defocus blur    | 25.0% | 25.0% |
> | Pixelate        | 45.3% | 49.2% |
>
>
> **Q1**: It does not tackle the problem from a perspective that I feel is significant. In particular, it is unclear to me how useful certified robustness to semantic transformations will be, especially for non-resolvable transformations. What does it mean to be robust to rotational blur for attack range $||\alpha||_2 < 20$ for CIFAR-10, for example? Does this correspond to any guarantees on real datasets, or at least empirical improvements on benchmark datasets?
>
> **A1**: The certified robustness against these non-resolvable semantic transformations means that you could not use **any** adaptively attack to fool the model within a given range. For example, rotational blur within $||\alpha||_2 < 20$ means an image is applied by a rotational blurring kernel averaged by rotations less than 20 degrees. It could be viewed as a subset of CIFAR-10-C mentioned in Dan Hendrycks et al (2019).
> The results mean that our models are **provably correct** on some samples in the dataset. This is a zero-to-one attempt compared with existing empirical defenses on these datasets. As for empirical experiments, we have conducted experiments for our method under empirical adaptive attacks and the key results are reported in the table above.
>
> **Q2**: However, their method usually performs very similarly or slightly worse than prior works when comparisons are possible; thus, the main experimental improvement is that it is now possible to achieve results on certain non-resolvable transformations that prior works are incapable of handling.
>
> **A2**: Yes. We admit that for some simple semantic transformations, we do not outperform the state-of-the-art. We suspect that this is because their bound on some transformations like rotation and scaling are tighter than ours. We would emphasize that our contribution is to extend its applications to complex semantic transformations like rotational blur rather than a tighter bound for randomized smoothing itself.
>
> **Q3**: How can we be sure that the generative model is capturing all the possible non-resolvable transformations that we may care about?
>
> **A3**:Thanks for your question. We think that
> 1. Can we learn all mathematical possible transformations?
>
> Probably not, you can easily construct some transformations that behaves like Dirichlet functions and they could not be fitted by neural networks. But these transformations do not appear in real life and any defense, no matter empirical or certified against these transformations will fail. This is phenomenon is relavant to the trade-off between accuracy and robustness discussed in prior works like (Hongyang Zhang et al.).
>
>
> 2. Can we learn practical semantic transformations effectively?
>
> Yes. Real semantic transformations are continuous and they do not harm the semantic features, at least for human beings. Thus neural networks are very effective when approximating them. Additionally, experiments also show that for many semantic transformations in real life, we could use NN to fit them.
>
>
> Therefore, we argue that we provide a new insight by using a surrogate model to study complex semantic transformations. Then we found both theoretical and empirical evidence that supports our idea. Although this might not solve the problem completely, at least we provide a solution for certifying against a broad range of semantic transformations in real datasets like some of CIFAR-C.
>
> **Q4**: Additionally, the paper could be presented better; in particular, the important components should be explained more clearly , whereas other parts of the paper can be moved to the Appendix. Can the notation section be simplified any further?
>
> **A4**:  Thanks for your advice on the writings of the paper. We have added more explanations and intuitive ideas on why our methods work for non-resolvable semantic transformations in revised revisions. We will try to use better notations to make the theorem and proof more friendly to readers in the revised version.

---

> > ### Author Response · Authors · 2021-11-11
> > **Thank you for your valuable review (Part 2)**
> >
> > **Q5**: Additionally, I believe the authors are missing very relevant prior work.
> >
> > **A5** : Thanks for the highly relevant works you mentioned. We have carefully read them and added them to the references. This paper present a general framework called model-based robustness and uses generative models to learn natural variation. We also adopt a similar idea by using a surrogate neural network to learn semantic transformations. But the difference is that our goal is to provide **certified radius** for these semantic transformations while the references study the **empirical robustness**.
> >
> > **Q6**: How large is the error term epsilon in Theorem 3 look for the generative models used in the experiments?
> >
> > **A6**: Due to the strong representation ability of the proxy networks, the error for fitting a semantic transformation is usually less than 1e-4, and the term about the norm of $M^\ast$ is about 1$\sim$10 in experiments. So the total error term is less than 1e-3 in practice and we do not need to worry about this term. We will provide more detailed results in the revision.
> >
> > References:
> >
> > 1. https://arxiv.org/abs/1903.12261 Benchmarking Neural Network Robustness to Common Corruptions and Perturbations
> > 2. https://arxiv.org/abs/1901.08573 Theoretically Principled Trade-off between Robustness and Accuracy

---

> > > ### Comment · Reviewer_oUek · 2021-11-18
> > > **I appreciate your response - I have just 1 follow-up question**
> > >
> > > Hi - thank you for your response. I just have one follow-up questions, since I know it is closer to the deadline now. Could you present empirical robustness results, not on your own PGD+EoT attacks, but on a common benchmark like CIFAR-C?
> > >
> > > Again, I am asking because, we do not have clear evidence (e.g. an MTurk experiment with humans) that the semantic-perturbation-NN is capturing relevant semantic transformations effectively; thus, I think one good way to show that the semantic-perturbation-NN is capturing important semantic transformations is simply to show that training to be (certifiably) robust to such NN-induced-perturbations will also lead to empirical robustness on a common benchmark like CIFAR-C.

---

> ### Author Response · Authors · 2021-11-26
> **A gentle reminder to reviewers**
>
> Dear reviewer oUek
>
> Thank you again for your valuable comments and suggestions, which are really helpful for us. We have posted responses to the detailed concerns.
>
> We totally understand that this is a quite busy period, since the reviewers may be responding to the rebuttal of other assigned papers.
>
> We deeply appreciate it if you can take some time to return further feedback on whether our responses solve your concerns. If there are any other comments, we will try our best to address them.
>
> Best,
>
> The authors

---

### Author Response · Authors · 2021-11-11
**Thanks for the valuable reviews and comments. A revised version of our paper is uploaded.**

We thank you again for all your valuable reviews and advice. We have uploaded a new version of our paper.  The main modifications are listed below

1.  We revised the presentation of our paper, especially **Section 3 the central idea and explanations of our GSmooth**. We also correct typos and **check the dimensions when introducing the data and noise augmentation** to make them more friendly to readers.

2. We have updated new experiments in Appendix 3. We test our model under **adaptive attack** to show the empirical accuracy and the tightness of the certified bound. The experiments are conducted on the CIFAR-10 dataset and we use projected gradient descent (PGD) by using expectation over transformation (EOT) to calculate the gradient for semantic transformations. The results are consistent with our theoretical analysis.

---

### Author Response · Authors · 2021-11-21
**A revised version of our paper is uploaded**

Dear all reviewers,

Thanks for your valuable reviews and questions. We have uploaded a new version of our paper.  The modifications compared with the last version are highlighted in blue color.


1. We update the recent experimental results and add them in Appendix 1.

2. We revised the writings of parts of the introduction and method and emphasize the intuition of our method to make it easier to understand.

Looking forward to your reply!

---

### Author Response · Authors · 2021-11-29
**Response to AC**

We thank you and all reviewers for your great efforts in handling our paper.

**Pros**

The reviewers acknowledge the novelty of our idea and methods of using the surrogate model for certifying semantic transformations and decoupling resolvable parts and non-resolvable parts of semantic transformations.

**Cons**

The major concerns of our paper are the lack of empirical results on common benchmarks like CIFAR-10-C to verify the tightness of the bound, and the results of several baselines are absent.

**Response**

1. To address these concerns, we have conducted lots of experiments on common benchmarks like CIFAR-10-C as well as the empirical adaptive attack using PGD under EOT for semantic transformations to demonstrate the empirical robustness of our method.

2. We have re-run the baselines pointed out by the reviewer. All these results are updated in the paper.

3. We also list a table that compares the limitations and methods of all baselines we compared in the paper.

In the revision, we have provided additional results and corrected some typos. We will try our best to further improve our paper in the paper and we deeply appreciate it if you could accept this paper.

---

### Decision · Program_Chairs · 2022-01-20

**Decision:**

Reject

**Comment:**

This paper proposes a more generalized form of certified robustness and attempts to provide new results on applying randomized smoothing to semantic transformations such as different types of blurs or distortions. The main idea is to use an image-to-image neural network to approximate semantic transformations, and then certify robustness based on bounds on that neural network. The authors provide empirical results on standard benchmark datasets like MNIST and CIFAR showing that their method can achieve improved results on some transformations compared to prior work.

The review committee appreciates the authors taking the time to attempt to respond to the concerns of all reviewers, and for updating and improving their work during the rebuttal process. The committee is glad to see that they do provide empirical evidence of improvement to common-corruption robustness, compared to AugMix (one of the state-of-the-art approaches for standard common-corruption robustness) and TSS.


However, the reviewers still have concerns about the novelty of the paper. The main novelty is not improvement for resolvable transformations (prior works that the authors cite perform about the same or better), but rather, is the ability to handle non-resolvable transformations. The reviewers agree that robustness to non-resolvable transformations is important; however, the reviewers think certified robustness to non-resolvable transformations is not meaningful, because they are only being certified with respect to a neural network that is trained to approximate those non-resolvable transformations. Without MTurk studies to confirm how good the neural network's non-resolvable transforms are, the reviewers do not find certified robustness here meaningful.